

# Biogeochemical fluxes and fate of diazotroph derived nitrogen in the food web after a phosphate enrichment: Modeling of the VAHINE mesocosms experiment

Audrey Gimenez[1], Melika Baklouti[1], Sophie Bonnet[1,2], and Thierry Moutin[1]

[1]Aix Marseille Université, CNRS/INSU, Université de Toulon, IRD, Mediterranean Institute of Oceanography (MIO) UM110, 13288, Marseille, France
[2]Institut de Recherche pour le Développement (IRD), AMU/CNRS/INSU, Université de Toulon, Mediterranean Institute of Oceanography (MIO) UM 110, 13288, Noumea, France-New Caledonia

*Correspondence to:* A. Gimenez (audrey.gimenez@mio.osupytheas.fr)

**Abstract.** The VAHINE mesocosm experiment in the oligotrophic waters of the Noumea lagoon (New Caledonia) aimed to assess the role of the nitrogen input through $N_2$ fixation on carbon production and export, and to study the fate of diazotroph-derived nitrogen (DDN) throughout the planktonic food web. A 1D-vertical biogeochemical mechanistic model was used in addition to the *in situ*
experiment to complement our comprehension of the dynamics of the planktonic ecosystem and the main biogeochemical carbon (C), nitrogen (N), phosphate (P) fluxes. The mesocosms were intentionally enriched with 0.8 $\mu$mol.L$^{-1}$ of P to trigger the development of diazotrophs and amplify biogeochemical fluxes. Two simulations were run, one with and the other without the phosphate enrichment. In the P-enriched simulation, $N_2$ fixation, primary production and C export increased
by 201, 208 and 87 %, respectively, consistent with the observed trends observed in the mesocosms (+ 124 %, + 141 %, + 261 % for $N_2$ fixation, PP and C export, respectively). The increase in primary and export productions became significant 10 days after the DIP enrichment, indicating that i) several days were necessary to obtain a significant response at the population scale, and ii) classical methods (short-term microscosms experiments) used to quantify nutrient limitations of primary
production may not be relevant. The model allowed to follow the fate of fixed $N_2$ by providing over time the proportion of DDN in each compartment (mineral and organic) of the model. At the end of the simulation (25 days), 43 % of the DDN was found in the non-diazotroph organisms, 33 % in diazotrophs, 16 % in the dissolved organic nitrogen pool, 3 % in the particulate detrital organic pool and 5 % in traps, indicating that $N_2$ fixation efficiently benefitted to non-diazotrophic organisms and
contributed to C export.



## 1 Introduction

Dinitrogen ($N_2$) fixation is the major external source of new nitrogen (N) for the upper ocean (Gruber
and Galloway, 2008; Mahaffey et al., 2005) and particularly in the south-western Pacific ocean
(Raimbault and Garcia, 2008; Moutin et al., 2008), which is recognized as one of the highest $N_2$
fixation area in the global ocean (Luo et al., 2012; Bonnet et al., 2015). While N availability primarily
controls autotrophic plankton growth in low nutrient low chlorophyll (LNLC) ecosystems (Moore
et al., 2001a; Graziano et al., 1996), the new N sources provided by $N_2$ fixation may drive the
planktonic ecosystem from N limitation toward P limitation and may potentially affect the magnitude
of C fixation, and eventually C export through the so called $N_2$-primed prokaryotic C pump (Karl
et al., 2003, 2012). Consequently, it is important to quantify N fluxes to the ocean as well as to study
the fate of N newly fixed by diazotrophs (or diazotroph-derived N, hereafter referred to as DDN) to
understand how $N_2$ fixation affects nutrient cycles and productivity in the ocean.

Biogeochemical models including $N_2$ fixation have been developed over the last decades, but
only a few of them have included diazotrophic organisms as state variables (Rabouille et al., 2006).
In these models, *Trichodesmium* is the most represented organism since it is the most studied dia-
zotroph and its physiology is well documented in the literature (Moore et al., 2001b; Fennel et al.,
2001; Moore et al., 2004; Rabouille et al., 2006). In recent studies, other diazotrophs such as uni-
cellular Cyanobacteria (termed UCYN) or diatom-diazotroph associations (termed DDA) have been
implemented in biogeochemical models. This was first done by Goebel et al. (2007) who developed a
diagnostic model to assess the relative contribution of three distinct diazotrophs (i.e. *Trichodesmium*
sp. and two UCYN from Group A and Group B (UCYN-A and UCYN-B, respectively) at the trop-
ical North Pacific station ALOHA. More recently, other biogeochemical models including a more
complex planktonic food web and the contribution of *Trichodesmium* sp., UCYN-A and UCYN-B
were developed (Monteiro et al., 2010, 2011), together with models representing *Trichodesmium* sp.,
and a general group of UCYN (Dutkiewicz et al., 2012). Although more and more models include
diazotrophic organisms as part of the food web, none of them have yet focused on the fate of DDN
throughout the ecosystem. Diazotrophs release a part of the recently fixed $N_2$ as dissolved organic
N (DON) and ammonium ($NH_4^+$) in the dissolved pool (Glibert and Bronk, 1994; Mulholland et al.,
2006). The magnitude of this release (10 to 80 %) is still under debate in the scientific community
(Glibert and Bronk, 1994; Konno et al., 2010; Benavides et al., 2013b, a) and seems to depend on
the physiological state of the cells (Berthelot et al., 2015a) as well as on exogenous factors such as
viral lysis (Hewson et al., 2004) or sloppy feeding (O'Neil et al., 1996). Nevertheless, recent meth-
ods coupling $^{15}N_2$ isotopic labeling, cell sorting by flow cytometry and high-resolution nanometer
scale secondary ion mass spectrometry (nanoSIMS) analyses recently allowed to quantify the DDN
transfer from diazotrophs to specific groups of non-diazotrophic phytoplankton and bacteria, indi-
cating that the DDN released in the dissolved pool is available and actively used by surrounding
non-diazotrophic communities (Bonnet et al. (a),subm., this issue).





The VAHINE project aimed to investigate the fate of DDN in oligotrophic ecosystems by deploy-
ing large-volume ( 50 m$^3$) mesocosms to isolate a water mass affected by a diazotroph development,
and by combining both field biogeochemical/diversity measurements and a mechanistic modeling
approach. The New Caledonian (Noumea) lagoon is considered as an oligotrophic ecosystem influ-
enced by oceanic waters inflowing from outside the lagoon (Ouillon et al., 2010). It supports high N$_2$
fixation rates (235 $\mu$molN.m$^{-2}$.d$^{-1}$, Garcia et al. (2007)), high *Trichodesmium* sp. (Dupouy et al.,
2000; Rodier and Le Borgne, 2008, 2010) and UCYN abundances (Biegala and Raimbault, 2008).
This site therefore represented an ideal location to investigate the fate of DDN.

The mesocosms were intentionally enriched with dissolved inorganic phosphate (DIP) to enhance
the potential development of N$_2$ fixers in the mesocosms, and therefore amplify N$_2$ fixation fluxes
and facilitate the study of DDN pathways in the planktonic ecosystem. Complementary field ap-
proaches were used during the VAHINE project including a $\delta15N$ budget to assess the dominant
source of N (from NO$_3^-$ - and/or N$_2$ fixation) fueling export production along the experiment (Knapp
et al., subm., this issue). Bonnet et al. (a) (subm., this issue) explored the fate of DDN at shorter
time scales, investigating the relative contribution of each diazotroph phylotype to direct C export,
and quantifying the DDN release and its subsequent transfer to different groups of plankton by using
nanoSIMS. In the present study, we developed a 1D-vertical biogeochemical model including the
representation of *Trichodesmium* and UCYN diazotrophs of the Group C (UCYN-C, which devel-
oped extensively during the mesocosm experiment, (Turk-Kubo et al., subm., this issue)). The goal
of this study was to complement our comprehension of the dynamics of the planktonic food web and
the associated biogeochemical fluxes during the mesocosm experiment by providing information
that could not be inferred through *in situ* measurements. We also used the model to follow the route
of DDN into the different compartments of the ecosystem (diazotrophs, non-diazotrophs, dissolved
pool, detrital pool and export).

## 2   Methods

### 2.1   The VAHINE experiment

The VAHINE experiment took place in January-February 2013 (austral summer) in the oligotrophic
New Caledonian lagoon. Three large volume (∼50 m$^3$, 15 m-height) mesocosms equipped with
sediment traps fixed at their bottom were deployed and the dynamics of the three mesocosms was
followed for 23 days. The full description of the mesocosms design and deployment, including
selection of the study site and logistics are detailed in Bonnet et al. (b) (subm., this issue). The
mesocosms were enriched with ∼0.8 $\mu$mol.L$^{-1}$ of orthophosphate (PO$_4^{3-}$) on the evening of day 4
to alleviate any potential DIP limitation which is a constant feature observed in the south-western
Pacific (Moutin et al., 2005, 2008) and stimulate N$_2$ fixation. Seawater was sampled daily in the
three mesocosms (hereafter called M1, M2 and M3) and outside (hereafter called lagoon waters) at





three depths (1m, 6m and 12m) and the sediment traps were collected every 24 h by scuba divers. The methods used to measure the different variables (C, N, P pools and fluxes, chlorophyll a stocks and plankton abundances) used in the present paper for comparison with the model simulations are detailed in the companion papers of Berthelot et al. (2015b), Leblanc et al., subm.,Van Wambeke et al., subm., this issue).

### 2.2 Mesocosm modeling and hypothesis

The model used in the VAHINE project is embedded in the modular numerical tool Eco3M (Baklouti et al., 2006), which uses mechanistic formulations to describe the biogeochemical processes engaged in the dynamics of marine pelagic ecosystems. Eco3M provides high flexibility by allowing its users to remove or add variables or processes to better adapt the model to a specific study. The VAHINE experiment consisted in the deployment of three replicate mesocosms in New Caledonia. Each mesocosm was modelled through a 1D box model with 14 boxes of 1m height each. Mass transfer between boxes is only allowed through sinking of particulate matter. Until day 10, only the detrital particles are allowed to sink but after this date, 10 % of all the living and non-living dissolved and particulate compartments are allowed to sink. This aims at representing the setting of the aggregation process and the subsequent intensification of the sinking process. The aggregation process was indeed supposed to be favored, not only by the reduced eddy fluxes due to the containment of water, but also by the release of TEP (Berman-Frank et al., subm., this issue). At the bottom of the modelled mesocosms, the sinking material is cumulated to be compared with the particulate matter collected daily in the traps. Sinking velocities were not measured during the experiment, and the matter daily collected in traps was used to parameterize the sinking velocity . The latter is therefore set constant until day 10 at 0.7 m.day$^{-1}$ and increases through the polynomial function given by equation (1) to reach 10 m.day$^{-1}$ at the end of the simulation :

$$V = \alpha * t^{10} + \beta \tag{1}$$

$$\alpha = \frac{(V_{max} - V_{min})}{t_{end}^{10} - t_{ini}^{10}} \tag{2}$$

$$\beta = V_{min} - \alpha * t_{ini}^{10} \tag{3}$$

where V is the sinking velocity, V$_{min}$ and V$_{max}$ are respectively the minimum and maximum sinking velocities (0.7 and 10 m.d$^{-1}$), t is time, t$_{ini}$ is the moment at which sinking rate starts to increase (i.e. day 10) and t$_{end}$ is the final day of the run (i.e. day 25). The VAHINE data also revealed that the water column inside the mesocosms was well mixed (Bonnet et al. (b), subm., this issue), probably through natural convection at night. This feature is simply modeled through a vertical homogenization of every concentration once a day (at midnight), by imposing the vertically-averaged concentrations in each box. Light irradiance data from the nearest meteorological station (Noumea airport)



were used for the surface irradiance in the model, and a vertical gradient was simulated thanks to a classical Beer-Lambert law using the attenuation coefficient found in Morel (1988). When the to-
tal N an P pools ($N_{total}$ and $P_{total}$) were calculated with the model outputs and compared to those obtained *in situ*, a significant difference appeared regarding $P_{total}$, while the $N_{total}$ fitted well (data not shown). This gap was mainly due to a DIP concentration that was too high compared to data, indicating a non-total consumption by organisms (not shown). To deal with this DIP excess in the system, a loss of DIP was added to the model. The main hypothesis to explain this DIP loss with-
out a similar loss in DIN is the formation of a biofilm of $N_2$-fixing organisms on the walls of the mesocosms (see Knapp et al. (subm., this issue) for details and DIP consumption calculations by the biofilm). Based on Knapp et al. (subm., this issue) calculations, this loss was estimated to $10\,\%.\mathrm{d}^{-1}$ and was assumed to have no influence on primary, bacterial or export production.

### 2.3    The biogeochemical model

The biogeochemical model used in this study is based on the Eco3M-MED model used for the Mediterranean Sea (Alekseenko et al., 2014). The only modification made on this previous version lies in the addition of diazotrophs and $N_2$ fixation process to adapt the model to the VAHINE experiment. The model includes 8 Planktonic Functional Types (PFT): four primary producers (autotroph phytoplankton), three consumers (zooplankton) and one decomposer (heterotrophic bacteria). All of
them are represented in terms of several concentrations (C, N, P and chlorophyll concentrations for phytoplankton) and abundances (cells or individual per liter) (Mauriac et al., 2011). Phytoplankton is originally divided into two size-classes, namely the large phytoplankton ($\geq 10~\mu$m) (PHYL) and the small phytoplankton ($\leq 10~\mu$m) (PHYS). The two $N_2$-fixing organisms are also distinguished by their size, the large one representing *Trichodesmium* sp. (TRI) and the small one *Cyanothece*
sp. (UCYN-C), which highly developed in the mesocosms during the experiment (Turk-Kubo et al., 2015). The zooplankton compartment is also divided in the three size classes, nano-, micro- and mezo-zooplankton, which respectively represent heterotrophic nanoflagelates (HNF), ciliates (CIL) and copepods (COP). The latter is represented in terms of abundance and C, N, P concentration. This differs from the model described in Alekseenko et al. (2014), in which meso-zooplankton is only rep-
resented through an abundance and a C concentration. Three nutrients are considered, namely nitrate ($NO_3^-$), ammonium ($NH_4^+$) and phosphate (DIP). The dissolved organic pool (DOM) is composed of labile and semi-labile fractions of DOC (LDOC and SLDOC), and labile fractions of DON and DOP (LDON and LDOP). The refractory organic pools are not represented. Finally, the detrital particulate matter is represented in terms of C, N and P ($\mathrm{Det}_C$, $\mathrm{Det}_N$ and $\mathrm{Det}_P$). All the biogeochemical
processes and interactions between the state variables are described in Fig. 1. Except for the new parameters associated with the new features of the model as compared to the original one (Alekseenko et al., 2014), the common parameters between the two model versions are identical.



### 2.3.1 Initial conditions

Initial values for the model state variables were derived from the *in situ* measurements averaged
over the three mesocosms and the three sampling depths (1m, 6m, 12m). Measured DOM values
included the refractory organic matter while the model only represents the labile (and semi-labile for
C) fraction. To extract the labile fraction from DON data, we considered that the labile DON cor-
responded to the quantity consumed during the experiment in the mesocosms which was estimated
at 1 $\mu$mol.L$^{-1}$. The percentage of the labile portion over the total DON was calculated and then ap-
plied to DOP to estimate the initial concentration of labile DOP. The available DOC fraction (LDOC
+ SLDOC) was evaluated at 5 $\mu$mol.L$^{-1}$ in the Equatorial Pacific (Pakulski and Benner, 1994).
PHYL was initialized with diatoms data, and PHYS with the sum of nanoEukaryotes, picoEukary-
otes, *Synechococcus* sp. and *Prochlorococcus* sp.. The initial detrital particulate matter was derived
by substracting the total living particulate matter considered in the model to the total particulate mat-
ter measured *in situ*. Accounting for the lack of data of zooplankton, we initiated the variables using
different ratios between BAC, HNF and CIL abundances. HNF$_{cell}$ was 50 times less than BAC$_{cell}$
and CIL$_{cell}$ was 2500 times less than HNF$_{cell}$. The standard value of 0.5 ind.L$^{-1}$ was used for adult
COP$_{ind}$ which is consistent with the recent results of Hunt et al. (subm., this issue). The initial values
of C, N, P and chlorophyll concentrations for the planktonic compartments were derived from the
initial cellular abundance data and from arbitrarily fixed intracellular contents (table 1).

### 2.3.2 Modeling N$_2$ fixation

The mathematical formulation (see Eq. (4)) used to represent N$_2$ fixation was adapted from Rabouille
et al. (2006) in order to be compatible with the formalism of the present model. It describes the N$_2$
fixation flux as a function of the nitrogenase (i.e. the enzyme catalyzing N$_2$ fixation) activity (Nase)
and the diazotroph abundance (DIAZO$_{cell}$, where DIAZO either refers to TRI or UCYN-C). The
N$_2$ fixation flux is regulated by the intracellular C quota and the N:C and P:C ratios (equation (8))
and by the intracellular N quota and N:C ratio (equation (9)). Intracellular N quota controls the net
N$_2$ fixation rate through a quota function (1-f$_{Q_N}$, equation (9)), the N excess exuded being equally
distributed into the DON and NH$_4^+$ pools. As in Rabouille et al. (2006), the nitrogenase activity
(Nase, in molN.cell$^{-1}$.s$^{-2}$) is a state variable, which dynamics is described in equation (5). The
nitrogenase activity results from the balance between the increase and the decrease in its activity.
The increase in the potential nitrogenase activity is assumed to be controlled by the N intracellu-
lar quota (equation (6)) and by the NO$_3^-$ concentration in field (equation (7)). *Trichodesmium* are
non-heterocystous filamentous cyanobacteria with differentiated cells located in the center part of
the colony called diazocysts (Bergman and Carpenter, 1991), where N$_2$ fixation occurs. This spatial
segregation mechanism is used by the organism to protect the nitrogenase enzyme from oxygen inac-
tivation produced by photosynthesis (Carpenter and Price, 1976; Bryceson and Fay, 1981). Besides,




*Trichodesmium* combines a spatial and temporal segregation to maximize the protection of the nitrogenase. This therefore allows the cells to fix only a few hours in daytime around noon (Roenneberg and Carpenter, 1993; El-Shehawy, 2003; Berman-Frank et al., 2001)). On the contrary, UCYN-C can only use a temporal strategy to separate $N_2$ fixation and photosynthesis processes and thus need to fix $N_2$ to protect the nitrogenase from $O_2$, released by photosynthesis during the day (Reddy et al., 1993). The inhibition of $N_2$ fixation during the day for UCYN-C and during the night for TRI is simulated by the $f_{inhib}$ function (equation (10), 12 h lagged between TRI and UCYN-C) which controls the nitrogenase activity. The decrease in nitrogenase activity is regulated by a saturation function involving $Nase_{dec}^{max}$, a coefficient of nitrogenase degradation (see equation (5) and table 2 ). Both the increase and decrease in nitrogenase are energy dependent and controlled by the intracellular C quota (equation (8)).

$$\underbrace{Flux_{N_{2_{fix}}}}_{molN.l^{-1}.s^{-1}} = \underbrace{Nase}_{molN.cell^{-1}.s^{-1}} \times \underbrace{DIAZO_{CELL}}_{cell.l^{-1}} \times f_{Q_C} \times (1 - f_{Q_N}) \tag{4}$$

$$\frac{dNase}{dt} = \underbrace{\overbrace{Nase_{prod}^{max}}^{\text{Maximum rate of increase}} \times min(f^{Nase}, f^{NO_3^-}) \times f_{Q_C} \times f_{inhib}}_{\text{Increase in Nitrogenase activity}} - \underbrace{\overbrace{Nase_{dec}^{max}}^{\text{Maximum rate of decrease}} \times \frac{Nase}{Nase + K_{Nase}} \times f_{Q_C}}_{\text{Decrease in Nitrogenase activity}} \tag{5}$$

$$f^{Nase} = \min\left( \max\left[ \left( \frac{Q_{NC}^{max} - Q_{NC}}{Q_{NC}^{max} - Q_{NC}^{min}} \right)^{0.06}, 0 \right], \left( \frac{Q_N^{max} - Q_N}{Q_N^{max} - Q_N^{min}} \right)^{0.06}, 1 \right) \tag{6}$$

$$f^{NO_3^-} = \frac{1}{1 + \frac{NIT}{K_{NO_3^-}}} \tag{7}$$

$$f_{Q_C} = \max\left( \left[ \left( \frac{Q_{NC}^{max} - Q_{NC}}{Q_{NC}^{max} - Q_{NC}^{min}} \right)^{0.06}, 0 \right], \max\left[ \left( \frac{Q_{PC}^{max} - Q_{PC}}{Q_{PC}^{max} - Q_{PC}^{min}} \right)^{0.06}, 0 \right], \left( \frac{Q_C - Q_C^{min}}{Q_C^{max} - Q_C^{min}} \right)^{0.06}, 1 \right) \tag{8}$$





$$
\quad f_{Q_N} = \begin{cases} 0 \text{ si } Q_N \leq Q_N^{min} \\ \min\left\{ 1 + \left| \dfrac{Q_N^{max} - Q_N}{Q_N^{max} - Q_N^{min}} \right|^{0.06}, 2 \right\} \text{ if } Q_N \geq Q_N^{max} \\ 1 - \left( \dfrac{Q_{NC}^{max} - Q_{NC}}{Q_{NC}^{max} - Q_{NC}^{min}} \right)^{0.06} \text{ if } Q_N \in [Q_N^{min}, Q_N^{max}] \text{ and } Q_{NC} \leq Q_{NC}^{max} \\ \min\left\{ 1 + \left| \dfrac{Q_{NC}^{max} - Q_{NC}}{Q_{NC}^{max} - Q_{NC}^{min}} \right|^{0.06}, 2 \right\} \text{ else} \end{cases} \quad (9)
$$

$$
f_{inhib} = \exp(3.7(cos(2\pi t - \pi) - 1)) \quad (10)
$$

### 2.3.3 Parametrization of diazotrophs and diazotrophs activity

*Trichodesmium* sp. and unicellular cyanobacteria (Group C and specially *Cyanothece* sp.) exhibit distinct physiologies, sizes and morphologies. Regarding the parametrization of diazotrophs and the
processes they undertake that are common with non-diazotrophs, it has arbitrarily been considered that *Trichodesmium* cells are equivalent to PHYL cells and the TRI state variable was therefore parameterized like 100 PHYL cells (considering that a trichome includes 100 cells, Luo et al. (2012)), and UCYN-C were parameterized like PHYS. For the diazotrophy process, parameters for TRI were configured using the Rabouille et al. (2006) work. TRI was also hypothesized to be not grazed in
the field. Its main predator is the copepods from the Harpacticoida order (mostly Macrosetella and Miracia) (O'Neil and Roman, 1992), which are not significantly found in the study area (Hunt et al., subm., this issue). To our knowledge, Grimaud et al. (2013) was the first to propose a dynamical model to depict the N$_2$ fixation by a unicellular cyanobactera (UCYN-C, *Crocosphaera watsonii*). Nevertheless, since this formulation of N$_2$ fixation was different from that of Rabouille et al. (2006),
we couldn't use the parameters provided in Grimaud et al. (2013). The latter were therefore derived from that of TRI not only on the basis of cell size considerations, but to obtain a global agreement with N$_2$ fixation fluxes measured during the experiment. All the parameters added for both TRI and UCYN-C new compartments are detailed in table 2.

### 2.4 The fate of N$_2$ fixed

The main purpose of the DIP enrichment was to enhance diazotrophy in the mesocosms and facilitate the measurement of the DDN transfer. To follow the pathways of DDN throughout the food web, a post treatment was realized since the model itself does not allow to distinguish between the DDN and other N sources. The post-processing treatment aimed at dynamically calculating the DDN proportion in each compartment of the biogeochemical model . At the beginning of the simulation,
we assumed that DDN was equal to zero in each compartment. We further assumed that the ratio DDN/N in each N flux leaving a given compartment was the same as the one within this compartment. DDN transfer starts with N exudation by diazotrophs. This DDN release fueled the DON and



$NH_4^+$ compartments, which are then taken up by autotrophs and heterotrophs. Grazing by zooplankton on the lower trophic levels will then transfer part of the DDN by excretion, sloppy feeding, and

egestion of fecal pellets. Finally, remineralization and natural mortality will also contribute to the transfer of DDN among the planktonic food web. Fig. 2 illustrates the different processes involved in the DDN transfer within the ecosystem.

## 3   Results

Two simulations of the mesocosm experiment were run: the first one includes the representation of

the DIP enrichment ($SIM^E$), while the second one does not consider this enrichment ($SIM^C$). The latter simulation can be considered as a proxy of the planktonic dynamics outside the mesocosms in lagoon waters. For the sake of clarity and better readability, prefixes "m" and "o" will be used to refer to model and observations respectively, and a * will be used to notify data measured outside the mesocosms. A vertical homogeneity was observed in the mesocosms during the experiment for most

of the biogeochemical and diversity parameters (Berthelot et al. (2015b), Turk-Kubo et al., subm., Leblanc et al., subm., this issue) . We thus used the average of the three sampling depths to plot both model results and observations. Three periods (namely P0, P1 and P2) were distinguished during the experiment based on biogeochemical characteristics (Berthelot et al. (2015b), this issue) and on changes in the diazotroph community composition (Turk-Kubo et al., subm., this issue). P0 stands

for the few days before the DIP enrichment, P1 for the period just after the enrichment (i.e. from day 5 to day 14), and P2 for the period from day 15 to day 23.

### 3.1   Dynamics of the different N and P pools

During P0, mDIP in $SIM^E$ decreases slowly from 47 to 24 nmol.L$^{-1}$ (Fig. 3, (a)). In response to the DIP enrichment at the end of day 4, mDIP reaches 830 nmol.L$^{-1}$, before gradually decreasing

to the low concentrations observed before the enrichment (Fig. 3, (a)). During the experiment, the DIP enrichment led to three different oDIP in the 3 mesocosms with 740, 780 and 990 nmol.L$^{-1}$ in M1, M2 and M3, respectively, reflecting the slightly different volumes of the mesocosms (Bonnet et al. (b), subm., this issue). oDIP then decreased below the quantification limit of 50 nmol.L$^{-1}$ in the three mesocosms but the consumption of oDIP in M1 was the fastest and those in M2 the slow-

est. Without the DIP enrichment ($SIM^C$), mDIP is quickly consumed and the concentrations remain close to zero until the end of the simulation, consistent with oDIP* which was < 50 nmol.L$^{-1}$ all along the experiment.

As for $oNH_4^+$, $nmNH_4^+$ remains low and stable around 15 nmol.L$^{-1}$ over the experiment (not shown here). $mNO_3^-$ also fits well with $oNO_3^-$, with nearly constant concentrations close to the quantifica-

tion limit of 50 nmol.L$^{-1}$ (Fig. 3, (b)) over the whole experiment.

oDOP and oDON remained relatively stable throughout the experiment with values around 5 $\mu$mol.L$^{-1}$



and 0.14 $\mu$mol.L$^{-1}$, respectively, with a slight decrease in P2 at the end (Fig. 3, (c) and (d)). A slight increase in mDOP in SIM$^E$ and a slight decrease in both mDOP and mDON are observed during P2. For SIM$^C$, mDOP and mDON remain stable throughout the simulated period (Fig. 3, (c) and (d)).

The trend was similar for oPOP and oPON with constant concentrations or a slight decrease during P1, followed by a large increase (by a factor of 1.5, 1.5 and 2 in M1, M2 and M3, respectively, in oPON, and by a factor of 1.4, 1.4 and 2.4 in M1, M2 and M3, respectively, in oPOP ) during P2 (Fig. 3, (e) and (f)). SIM$^E$ results are in good agreement with data for mPON which starts at 1 $\mu$mol.L$^{-1}$ and then increases to a maximum of 1.5 $\mu$mol.L$^{-1}$ during P2. While oPOP decreased

slightly at the beginning of P1 and increased during P2, mPOP in SIM$^E$ remains constant (0.08 $\mu$mol.L$^{-1}$) from day 5 to 10 and increases after day 10. The increase in mPOP up to the 0.14 $\mu$mol.L$^{-1}$ peak is higher and occurs earlier than oPOP, before to decrease as in the observed data at the end of P2. In SIM$^C$, the total particulate organic matter falls down throughout the entire simulation, from 0.06 to 0.02 $\mu$mol.L$^{-1}$ for mPOP, and from 1 to 0.4 $\mu$mol.L$^{-1}$ for mPON.

oN$_{total}$ averaged 6.2 $\mu$mol.L$^{-1}$ during P1 and started to decrease at the end of P2 in the mesocosms (Fig. 3, (h)). mN$_{total}$ in both SIM$^E$ and SIM$^C$ are quite similar and in the same range as the one observed in data, with a decrease a bit stronger for SIM$^C$ at the end of P2. SIM$^E$ shows an immediate and high increase in mP$_{total}$ (1-1.2 $\mu$mol.L$^{-1}$) on day 5, corresponding to the DIP enrichment, while mP$_{total}$ in SIM$^C$ is constant (250 nmol.L$^{-1}$) throughout the simulation (Fig. 3, (g)). After the

enrichment, mP$_{total}$ starts to decrease down to 0.2-0.25 $\mu$mol.L$^{-1}$ on day 22. oChl remained stable during P1 and increased during P2 by a factor of 5 up to a maximum of 1 $\mu$g.L$^{-1}$ in M3 (Fig. 4 (a)). oChl a was lower (0.6 $\mu$g.L$^{-1}$) in M1 and M2 at the end of P2. mChl a calculated by SIM$^E$ is similar to oChl a in M1 and M2, with a decrease a little more marked during P1 and a maximum of 0.5 $\mu$g.L$^{-1}$ in P2. When mChl a increases during P2 in SIM$^E$, mChl a in SIM$^C$ remains stable

($\sim$0.1 $\mu$g.L$^{-1}$) until the end of the simulation.

### 3.2 Dynamics of the different fluxes

The biogeochemical fluxes relative to the main processes like primary and bacterial productions (PP and BP), N$_2$ fixation (N$_2^{fix}$), turnover time of DIP (T$_{DIP}$) and particulate matter export fluxes (POC$_{exp}$, PON$_{exp}$ and POP$_{exp}$) have been calculated by the model and compared to the measured

ones (Fig. 4 (b) to (h)).

At the beginning of P0, oN$_2^{fix}$ as well as mN$_2^{fix}$ (both in SIM$^E$ and SIM$^C$) were about 17 nmolN.L$^{-1}$.d$^{-1}$ and declined gradually during P1 down to 10 nmolN.L$^{-1}$.d$^{-1}$. While mN$_2^{fix}$ in SIM$^C$ continues to decrease during P2, mN$_2^{fix}$ in SIM$^E$ increases during P2 by a factor of 4, consistent with oN$_2^{fix}$, and reaching a maximum of 42 nmolN.L$^{-1}$.d$^{-1}$ on day 23. PP and BP exhibited the same temporal

dynamics in both data and SIM$^E$ results. They first slightly decrease before the DIP enrichment, remaine stable during P1 and increase during P2 by a factor of 4.4 and 2.7, for PP and BP, respectively (Fig. 4, (c) and (e)). During P2, mPP (SIM$^E$) rises until 2 $\mu$molC.L$^{-1}$.d$^{-1}$, which is in the





range of the oPP measured in the three mesocosms. M3 exhibited higher values of oPP than those in M1 and M2 during P2 (around 4 $\mu$molC.L$^{-1}$.d$^{-1}$ on day 22). Even if mBP (in SIM$^E$ and SIM$^C$) starts at a higher rate than oBP measured in the 3 mesocosms, it decreases rapidly from day 2 to 4 to reach the *in situ* value before the enrichment. The increase in mBP from day 11 to day 17 in SIM$^E$ is a bit overestimated compared to data. BP then better fits the data measured at the end of P2, and especially in M3 (500 ngC.L$^{-1}$.h$^{-1}$). In SIM$^C$, the increase in mBP and mPP during P2 does not occur and these rates remain constant around 0.5 $\mu$molC.L$^{-1}$.d$^{-1}$ for mPP and 200 ngC.L$^{-1}$.h$^{-1}$ for mBP throughout the 25 days of the simulation. mBP values in SIM$^C$ are lower than the ones measured in the three mesocosms and consistent with the oBP values measured in lagoon waters (Fig. 4, (e)). T$_{DIP}$ is a relevant indication of DIP availability in the water column. After a slight decline of oT$_{DIP}$ during P0 to values lower than 1 day, T$_{DIP}$ increased dramatically up to 30 days (oT$_{DIP}$) and 21 days (mT$_{DIP}$) after the DIP enrichment. mT$_{DIP}$ then decreases linearly in SIM$^E$ as well as oT$_{DIP}$, in the 3 mesocosms. mT$_{DIP}$ in SIM$^C$ has the same trend as the oT$_{DIP}$ measured in the lagoon waters (Fig. 4, (g)).

The fluxes of exported matter POC$_{exp}$, PON$_{exp}$ and POP$_{exp}$ for C, N and P respectively are represented in Fig. 4, (d), (f) and (h) in terms of dry matter measured in the sediment traps (Berthelot et al., 2015b). During P1, the daily export remains relatively stable and averaged 18, 1.13 and 0.09 mmol for mPOC$_{exp}$, mPON$_{exp}$ and mPOP$_{exp}$, respectively. oPOC$_{exp}$, oPON$_{exp}$ and oPOP$_{exp}$ gradually increased during P2 (from day 15 to 25) to reach a maximum of 57 mmolC.d$^{-1}$, 5 mmolN.d$^{-1}$ and 0.5 mmolP.d$^{-1}$ respectively. In SIM$^E$, mPOC$_{exp}$, mPON$_{exp}$ and POP$_{exp}$ fit well data with a slight overestimation of mPOP$_{exp}$ at the end of P2 which reaches a maximum of 0.75 mmolP.d$^{-1}$. There is no significant difference between SIM$^E$ and SIM$^C$ for mC$_{exp}$ and mPON$_{exp}$, from the beginning of the experiment to the middle of P2 (day 18). From day 19, the increase in mPOC$_{exp}$ and mPON$_{exp}$ is less important in SIM$^C$ than in SIM$^E$. For mPOP$_{exp}$, the increase in SIM$^E$ occurs earlier (day 15) and the discrepancy between SIM$^E$ and SIM$^C$ is larger at the end of the simulation.

### 3.3 Evolution of planktonic abundances

The model also simulates the abundances of organisms in cell.L$^{-1}$ for single-cells, in trichome.L$^{-1}$ for TRI and in ind.L$^{-1}$ for zooplankton, besides being represented in term of biomass (C,N,P and Chl for phytoplankton).

mTRI remains constant in SIM$^E$ around 250 trichomes.L$^{-1}$. By constrast, a large development of UCYN-C occurs during P2 with mUCYN-C reaching 5.10$^7$ cell.L$^{-1}$ (Fig. 5, (a) and (b)). This increase in mUCYN-C is consistent with the observed dynamics, though the mUCYN-C increase is overestimated in SIM$^E$ compared to oUCYN-C. mPHYL decreases over time in both SIM$^E$ and SIM$^C$ (Fig. 5, (c)). In the three mesocosms, oPHYL increased from day 10 to 15 reaching 10$^5$ cell.L$^{-1}$ before decreasing back to values close to that of mPHYL. During P0, mPHYS decreases slightly like oPHYS. During P1, the decrease in mPHYS (down to 0.1 10$^8$ cell.L$^{-1}$) is stronger than



that of oPHYS, which increases since day 10 and reaches the same range of values as oPHYS at

the beginning of P2. During P2, mPHYS and oPHYS increase up to 1.5 $10^8$ cell.L$^{-1}$ for mPHYS in SIM$^E$ and 1.3-2.9 $10^8$ cell.L$^{-1}$ for oPHYS.While mPHYS are similar in SIM$^E$ and SIM$^C$ from day 2 to day 8, the increase in mPHYS after day 8 and until the end of P2 is lower in SIM$^C$ than in SIM$^E$ (Fig. 5, (d)). As for PHYS, there is a slight decrease in mBAC and oBAC during P0. The DIP enrichment on day 4 leads to a strong decline from day 5 to day 8, which is more marked in

mBAC (9.5 $10^7$ cell.L$^{-1}$ in SIM$^E$) than in oBAC (2.3-3.1 $10^8$ cell.L$^{-1}$). From day 8 to the end of the simulation, mBAC increases up to a maximum of 1.1 $10^9$ cell.L$^{-1}$, while oBAC reached a maximum of 6.8-8.5 $10^8$ cell.L$^{-1}$ at the end of the experiment. In the same way, mBAC are similar in SIM$^E$ and SIM$^C$ from day 2 to day 8, and then increase until the end of the simulation but to a lower extent in SIM$^C$ than in SIM$^E$. Since no zooplankton data that could be used for comparison

with the model results were available, only the dynamics of SIM$^E$ and SIM$^C$ are presented (Fig. 5, (g) to (f)). mHNF and mCIL have the same trends though they are time-shifted. mCOP is similar in SIM$^E$ and SIM$^C$, with a decline from 0.5 ind.L$^{-1}$ at the beginning to less than 0.1 ind.L$^{-1}$ at the end of the simulation. Except for mCOP and mPHYL, the DIP enrichment has an important impact on the plankton dynamics as significant differences between the results of SIM$^E$ and SIM$^C$ in mTRI,

mUCYN-C, mPHYS, mBAC, mHNF and mCIL are observed. Overall, SIM$^C$ presents abundances 3 to 680 times lower than those simulated by SIM$^E$, though the temporal trends were similar between the two simulations

### 3.4    DIP enrichment and diazotrophs growth

The model also gives additional information not provided by the data regarding the growth of the

organisms or about their intracellular contents. The population growth rate (in cell.L$^{-1}$.s$^{-1}$) as well as the specific growth rate (in s$^{-1}$) are plotted in Fig. 6 (a) and (b) while the relative intracellular C, N and P quotas (i.e. Q$_C$, Q$_N$ and Q$_P$) are plotted in Fig. 6 (c) and (d). The DIP enrichment at the end of day 4 has a direct impact on Q$_P$ for both TRI and UCYN-C, with an instantaneous increase in Q$_P$ up to 100 % on day 5. While Q$_C$, Q$_N$ and Q$_P$ for TRI remain at their maximum

value until the end, Q$_N$ and Q$_C$ of UCYN-C decrease as soon as Q$_P$ increases on day 5. During P2, Q$_P$ gradually declines for TRI and faster for UCYN-C. The reverse process then takes place with an increase in Q$_N$ and Q$_C$ for UCYN-C when Q$_P$ decreases since day 15, whereas this is not observed for TRI. All along the simulation, the trends of f$_{TRI_{trich}}^{growth}$ and f$_{TRI}^{growth}$ are similar, with a sudden increase on day 5 followed by rather constant and then decreasing values (Fig. 6 (b)). By

contrast, the increase in $_{UCYN-C_{cell}}^{growth}$ after the DIP enrichment (day 5) is not observed in f$_{UCYN-C}^{growth}$, namely at the population scale (Fig. 6 (a)). f$_{UCYN-C}^{growth}$ increases 10 days later, i.e.during P2, up to a maximum of 200 cell.L$^{-1}$.s$^{-1}$ on day 22.



### 3.5 Fate of DDN in the ecosystem

The fate of the N that has been fixed since the beginning of the simulation (DDN) has been examined
using the post-processing treatment described in the Methods section. In short, the proportion of the
total DDN present in each living and non-living compartment of the water column and in the traps
was calculated throughout the simulation period (Fig. 7). At the start of the experiment, DDN is
nearly exclusively in TRI (the proportion of DDN in UCYN-C is negligible) but this proportion
decreases all along the simulation. Until day 10, most of the DDN is transferred to the DON pool
(which contains about 35 % of the total DDN on day 10), followed by (in their order of importance)
$NH_4^+$ (up to 10 % on day 5), DET (12 % on day 10), and the components of the microbial loop.
Until day 10, the proportion of DDN in each compartment except TRI, either increases with time,
or reaches a maximum around day 5, consistent with the decrease in $N_2$ fixation rates during that
period. After day 10, the proportion of DDN increases in all living organisms, thereby indicating the
transfer of DDN to non-diazotrophic organisms. DDN proportion increases almost until the end of
the simulation in CIL and HNF, but only until day 18 in BAC, PHYS and MZOO before decreasing
again. In the non-living compartments, the proportion of DDN decreases after day 10 (day 12 for
DON) until the end of the simulation. Finally, the proportion of DDN in traps is almost null during
the ten first days of the experiment, before increasing and then stabilizing around 4%. The percentage
of DDN with respect to total particulate N contained in the traps has also been plotted (Fig. 8). This
percentage increases quite linearly with time from 0 to nearly 0.4 % between day 2 and day 10. On
day 10, the percentage increases much more rapidly until day 12 and then rises gradually to a plateau
around 1.2 % before increasing again at the very end of the experiment.

### 4 Discussion

N input by $N_2$ fixation in the upper SW Pacific Ocean is thought to be controlled by DIP availability
because of the presence of repleted trace metals concentrations compared to the adjacent South
Pacific central gyre Moutin et al. (2005, 2008)). The aim of the VAHINE experiment (Bonnet et al.,
b) (this issue) was to (i) investigate the fate of the DDN in oligotrophic ecosystems by removing any
potential DIP limitation for diazotrophs and thereby potentially stimulate the growth of organisms
(in particular diazotrophs), (ii) enhance $N_2$ fixation and DDN fluxes through the entire ecosystem,
and (iii) study the dynamics of biogeochemical C, N, P fluxes. $N_2$ fixation is expected to rapidly
deliver new N to other organisms than diazotrophs, thus reducing possible N growth limitation or
co-limitation in the ecosystem. Our goal was to follow the dynamics of this new N toward the food
chain, the inorganic and organic N pools, as well as in the exported particulate matter. The discussion
will concern expected and unexpected results obtained in this study after the DIP enrichment, as well
as the fate of DDN in the ecosystem.





### 4.1 An expected enhancement of biogeochemical fluxes after the DIP enrichment

The mesocosms DIP enrichment performed at the end of day 4 associated with the provision of new N by diazotrophy led to a large increase in diazotrophs (especially UCYN-C) abundances (Fig. 5, (a) and (b))), biomass (data not shown) and $N_2$ fixation fluxes during P2, and a significant development of UCYN-C occurred during that period. Whereas a large increase in $N_2$ fixation is observed in $SIM^E$ during P2 (consistent with the data indicating a near three-fold higher mean $N_2$ fixation rate in P2 than P1) (Fig. 4, (a)), $N_2$ fixation rates gradually decreases in $SIM^C$, indicating strong differences between the mesocosms conditions and those encountered in lagoon waters. Hydrological parameters such as temperature and biogeochemical conditions were all similar inside and outside the mesocosms, except the DIP conditions (Bonnet et al. (b),subm., this issue), confirming that the DIP enrichment stimulated $N_2$ fixation in this experiment. Nevertheless, a slight increase in $N_2$ fixation rates was observed outside the mesocosms during P2 (+35 %), which could be explained by a provision of external DIP sources to the lagoon, by growth on DOP sources (Dyhrman et al., 2006) and/or by the increasing seawater temperature along the 25-days experiment (Bonnet et al. (b),subm., this issue), which provides favorable conditions for diazotroph growth (Carpenter et al., 2004). A rapid decrease in $T_{DIP}$ was observed on day 5 after the DIP enrichment, suggesting a rapid consumption of the DIP by the planktonic community. Diazotrophs were the first to respond to the DIP enrichment in term of abundance, even if this response did not lead to an immediate increase in the $N_2$ fixation rate. The latter significantly increased during P2 in relation to the development of UCYN-C (Turk-Kubo et al., subm., this issue). Other autotrophic organisms and heterotrophic bacteria declined until the middle of P1, and started to grow 10 days after the DIP enrichment (except PHYL). Despite this time lag between the DIP enrichment and the planktonic response, the DIP enrichment resulted in an increase in the abundances of all planktonic groups except PHYL and COP (Fig. 5). The DIP limitation at the beginning of the experiment is represented in the model by setting the P cell contents of all organisms at their minimum value, which leads to an immediate uptake of DIP after the enrichment at the end of day 4 (Fig. 7, (c) and (d)). At the cellular scale, this immediate DIP uptake results in a fast increase in intracellular P-contents of autotrophs and heterotrophic bacteria up to their maximum quota (Fig. 7, (c) and (d) for diazotrophs). After benefiting diazotrophs, the DDN inputs benefited to non-diazotrophic organisms. Autotrophic PP and heterotrophic BP increased in the model after the DIP enrichment (+262 % and +181 %, from day 5 to day 23, for PP and BP, respectively). The enhanced PP (Fig. 5, (e)) leads to an increase in total suspended matter (Fig 3, (e) and (f)), and finally in exported particulate material (Fig. 5, (d), (f) and (h)). The contribution of $N_2$ fixation to PP (up to 10.0 % for $SIM^E$ and 6.0 % for $SIM^C$) is in good agreement with corresponding measured contributions which were equal to $10.9 \pm 5.0$ % inside the mesocosms and $5.7 \pm 2.0\%$ in the lagoon waters (Berthelot et al., 2015b). Hence, the DIP enrichment not only stimulates $N_2$ fixation and PP but also the percentage of PP sustained by $N_2$ fixation. The newly synthetized biomass has two possible fates, namely remineralization or export.



As the model does not represent the diatom-diazotroph associations (DDAs), which were the most abundant diazotrophs in the mesocosms during P1 (Turk-Kubo et al., subm., this issue), the modeled export is probably underestimated during P1. Berthelot et al. (2015b) (this issue) have shown that the growth of DDAs during P1 did not lead to a significant increase in POC because DDAs rapidly settled down(Villareal et al., 1996) through the water column and the DDN did not benefit to the system. This may explain why the export during P1 was lower than during P2, during which we observed a higher increase in suspended particulate matter (Fig. 5, (d), (f) and (h)) enhanced by the UCYN-C growth. Moreover, the presence of large (100-500 $\mu$m) UCYN-C aggregates in the mesocosms facilitated their export into the traps (UCYN-C accounted for up to 22.4 $\pm$ 4.0 %of total C export at the height of their large development (Bonnet et al. (a), subm., this issue). This indicates that UCYN-C can not only contribute to direct export but promote indirect export. The high content of TEP measured in traps on days 15 and 16 (Berman-Frank et al., subm., this issue) in correlation with the increase in UCYN-C abundances (Turk-Kubo et al., subm., this issue) leads to the assumption that the presence of TEP in the field would facilitate export flux and specially the sinking of UCYN-C during P2. This phenomenon was taken into account in the model by allowing, each day from day 10, the settling of 10 % of all the model compartments (living and non-living, particulate and dissolved) in addition to the detrital particulate matter. C, N, P export in SIM$^E$ closely follows the mesocosm trap measurements (Fig. 6, (d), (f) and (h)). SIM$^E$ shows higher C, N, P exports (+ 28 %, + 35 % and + 158 %, respectively) compared to SIM$^C$. Large size N$_2$-fixing organisms are known to directly contribute to C export in coastal and oceanic environments (Subramaniam et al., 2008; Karl and Letelier, 2008) but small-size UCYN-C (despite very few studies have focused on them) were considered as less efficient at promoting export due to their small size (typically 1-6 $\mu$m) associated with low individual sinking rates, and the tight grazing control that leads to high recycling rates in the euphotic zone. In the present study, both our experimental and model results indicate that UCYN-C also significantly contribute to export under DIP repleted conditions, both directly by the sinking of UCYN-C cells, and indirectly after the transfer of DDN to non-diazotrophic plankton, which is subsequently exported.

### 4.2 An unexpected delay for UCYN-C development and biogeochemical fluxes enhancement

The new N provided by N$_2$ fixation after the DIP enrichment resulted in high PP and BP rates, as well as in an increase in export and planktonic abundances. However, these responses were not observed immediately after the DIP enrichment on day 4 but 10 days later (Fig. 3 and Fig. 5). The massive UCYN-C development indeed occurred during P2, with a maximum population growth on day 21 in the model, consistent with the observation of the maximum in the UCYN-C abundances on days 20, 16 and 19 in M1, M2, and M3 respectively (Fig. 5, (b)) (Turk-Kubo et al., subm., this issue). Which factor may explain the 10 days delay between the DIP enrichment and the large UCYN-C development?



At the cellular scale (Fig. 6, (a)), the DIP enrichment has an immediate influence on cell-specific growth rate of UCYN-C, with a 4-fold increase in few hours. However, this immediate response is not observed at the population scale (Fig. 6, (a)). At the beginning of the simulation, P cell quota of UCYN-C is minimum and their cell specific growth rate is therefore equal to zero. Though DIP and DOP are very low at the beginning of the simulation, UCYN-C can however take up part of this

available P, thereby increasing their P quota and their growth rate. UCYN-C reach their maximum cellular P quota the day after the DIP enrichment (Fig. 6, (c)) and DIP will not limit the UCYN-C growth anymore until day 17. The peak in cell-specific growth rate at day 5 (Fig. 6, (c)) corresponds to the temporary absence of significant nutrient limitation, while oscillations during the following days correspond to the day/night rhythm in UCYN-C C quota associated with C starvation dur-

ing night (the specific growth rate is modulated by the lowest intracellular quota). When C is the most limiting nutrient, the night/day oscillations are passed on growth rate. The high increase in cell-specific growth rate on day 5 leads to an increase in UCYN-C abundances (Fig. 5, (b)). After day 5, photosynthesis and N uptake are then not rapid enough to sustain the increased C and N needs, and N, and mostly C at night, become limiting (Fig. 6, (c)). As a consequence, UCYN-C cell-

specific growth rate decreases slightly after day 5, and more rapidly after day 18 when DIP becomes once again limiting (Fig. 6, (a)). Fig. 6 illustrates the time lag between the variations at the cellular level for specific growth rate and growth at the population level. The growth rate of the UCYN-C population also increases from the beginning of the simulation since the specific growth rate and the abundance of UCYN-C increase, but this is almost imperceptible until the exponential increase

starting around day 11. From day 18, when the specific growth rate begins to strongly decrease, the population growth rate still increases but more slowly and finally decreases after the maximum of $5.10^7$ cell.$L^{-1}$.$s^{-1}$ reached on day 22 (Fig. 6, (a)). TRI abundance is less influenced by the DIP enrichment than UCYN-C abundance. However, the DIP enrichment leads to an increase in TRI growth rate on day 5 at both the population and the trichome scale (Fig. 6, (c)). Since a trichome

includes 100 cells of *Trichodesmium*, the time lag between the responses at the trichome and population levels is therefore far less than the one evidenced for UCYN-C. Furthermore, despite TRI growth is not nutrient-limited from day 5 to day 15 as the three cellular quotas (C, N and P) are at their maximum value (Fig. 6, (d)), TRI population does not increase significantly because of its low maximum division rate as compared to the time scale of the experiment (3 weeks) (consistently with

*in situ* data). The aforementioned time lag between cellular and population responses is also useful for understanding what may be viewed as a contradiction: on one hand, we observed a clear and net increase in PP, BP and export productions after the DIP enrichment, both in the mesocosms and in the SIM$^E$, but on the other hand, oligotrophic waters are generally known to be more DIN than DIP limited. After reviewing the main studies conducted on nutrients limitation, and especially on N

and P limitation in oligotrophic waters, Moore et al. (2013) concluded that N was the first limiting nutrient for phytoplankton in nutrient-depleted areas as nutrient-addition experiments did not lead




to a significant increase in autotrophic activity after P additions, whereas it did after N additions
(Thingstad et al., 2005; Moore et al., 2008; Tanaka et al., 2011; Zohary et al., 2005). Similar re-
sults were obtained in the South Pacific gyre for autotrophs (Bonnet et al. (2008)) and heterotrophs
(Van Wambeke et al., 2008), as well as at the start of the present mesocosm experiment (before the
DIP enrichment), confirming previous studies indicating proximal N limitation of BP (Van Wambeke
et al., subm., this issue) at short time scales (days). This apparent contradiction on DIP limitation
may therefore be explained by the time duration of the aforementioned DIP enrichment experiments
that was not long enough to evidence the response of the planktonic ecosystem. The enrichment
mesocosm experiment conducted during the VAHINE project has allowed to follow the ecosystem
and the associated biogeochemical fluxes over a longer period of time (23 days) compared to the
nutrient-addition experiments cited above. Since we observed the increase in PP and BP only after
10 days in both experimental and simulation results, we may conclude that 10 days are necessary
for the newly fixed N by diazotrophs to sustain the observed high production rates, and to see an
effective change in the planktonic populations (in term of abundances, structure and function). In
the light of the foregoing, two conclusions may therefore be drawn. First, 10 days may be a lower
time limit to characterize the real nutrient limiting primary, bacterial and export productions, at least
in marine areas where $N_2$ fixation is a significant process. Therefore, short-term ($\sim$2 days) nutrient-
addition experiments may not be relevant to study nutrient limitation in marine ecosystems. Second,
the initial DIP limitation considered in the model clearly indicates that DIP limitation observed at
the cellular level does not reflect the response at the population scale (in terms of primary, bacterial
and export productions) which may be delayed. Therefore, in order to correctly assess the nutrient
limitation during short-term nutrient addition experiments, nutrient limitation diagnostics operating
at cellular level (such as enzymatic responses) need to be applied rather than classical measurements
of PP or BP increase after the enrichment.

### 4.3 The fate of DDN in the planktonic ecosystem and exported matter

At the start of the simulation, DDN is nearly exclusively into TRI since the flux of $N_2$ fixation by
UCYN-C is negligible as compared to that of TRI, and the situation reverses at the end of the sim-
ulation when UCYN-C abundance becomes predominant. Due to DON exudation and $NH_4^+$ release
by TRI, the proportion of DDN first increases in the DON and $NH_4^+$ pools, and then in the $NO_3^-$ pool
due to nitrification. Before day 10, planktonic organisms do not significantly benefit from the DDN,
as its proportion decreases in BAC and PHYS between days 4 and 8 and in HNF between days 6 and
10. For BAC and PHYS, this is mainly due to the decrease (which is overestimated by the model)
in abundance of these both groups between days 5 and 8 due to grazing by HNF and CIL. After
day 10, the DDN proportion increases in all the non-diazotrophic plankton groups, and decreases
meanwhile in the non-living pools, though a bit later (i.e. from day 13) in DON. This decrease in
DDN proportion in the non-living pools is both due to the assimilation of mineral and organic nu-





trients by phyto and bacterioplankton, and to the sinking of the produced organic matter through aggregation processes. Since mineral N is first taken up, the uptake of DON occurs later, namely during P2 as shown in Berthelot et al. (2015b) (this issue). As a consequence, the decrease in DDN-DON percentage is also delayed as compared to that of $NO_3^-$ and $NH_4^+$. DDN-DET increases quite regularly until day 10 as long as the sinking rate is constant, and then decreases with the increase in this sinking rate. As a result, DDN in the particulate matter collected in traps increases since day 10 up to the end, consistent with the $\delta15N$-budget performed by Knapp et al. (subm., this issue), thereby indicating a higher contribution of $N_2$ fixation to export production during P2 (56 ± 24 % and up to 80 % at the end of the experiment) compared to P1 (47 ± 6 % and up to 60 %). mDON appears to be the pool which mainly benefits from the DDN. This is due to the DON release by diazotrophs, especially TRI which is at its maximum N quota throughout the simulation (Fig. 8, (c) and (d)). Since their maximum cell division rate is low, their $N_2$ fixation rate is indeed high enough to allow *Trichodesmium* to fulfill their N reserves and reach their maximum N quota (Fig. 6, (d)). The same is not true for UCYN-C for which the division rate (boosted by the P-enrichment) is too high, as compared to their N fixation rate, to reach their N maximum quota. However, in the model, DDN exudation by diazotrophs releases equal amounts of $NH_4^+$ and DON. During P1, DDN accumulates in DON (nearly up to 40 % on day 13) (Fig. 7, (c)) whereas DDN in $NH_4^+$ decreases rapidly since day 5 as it is immediately used by heterotrophic bacteria and phytoplankton (Fig. 7, (d)). DDN in DON decreases later (i.e. during P2, when the DON pool begins to be used) as the inorganic N pool is depleted. To conclude on this point, though DDN transits in the same proportions in $NH_4^+$ and DON, it mostly accumulates in DON since DDN-$NH_4^+$ is taken up more rapidly. Among the living compartments, PHYS, BAC, HNF and CIL were the main beneficiaries of DDN. PHYS and BAC are indeed the main consumers of $NH_4^+$ and labile DON (while PHYL is not allowed by the model to uptake DON), and HNF and CIL respectively feed on BAC and PHYS, and on PHYS and HNF. DDN therefore mainly transits through the actors of the microbial loop, which is consistent with NanoSIMS measurements performed after 24 h of incubation with $^{15}N_2$ on water sampled on day 17 showing that 18 ± 4 % of the DDN was found in picophytoplankton against 3 ± 2 % in diatoms (Bonnet et al., a). According to the model, only 5 % of the total DDN were recovered in the traps at the end of the simulation. This proportion is likely underestimated by the fact that UCYN-C sinking is probably underestimated in the model. The contribution of UCYN-C to POC export on day 17 during P2 was indeed 0.25 % in the model simulation, against up to 22.4 ± 4.0 % in the data during the same period (Bonnet et al. (a),subm., this issue). In the same way, the ratio DDN/total N in traps traps equals 1 % at the end of the simulation, which is dramatically lower than the measured one which is equal to 80 % (Knapp et al.,subm., this issue). This discrepancy is partially due to the different methodologies used to make these estimations. In the post-processing treatment, we indeed considered that the initial DDN was null in every compartment, which is obviously not true, but this hypothesis was constrained by the fact that the initial DDN in all the model compartments was un-



known, and arbitrary allocations of DDN in compartments would have added additional uncertainty on the model results. As a consequence, our results are necessarily underestimated as compared to the measured ones since the latter include the history of previous $N_2$ fixation in the field (i.e. before the beginning of the mesocosm experiment). If we consider an initial content of DDN in the traps equals 30 % as measured by Knapp et al. (subm., this issue), the final modeled DDN content would

be 31 %, which is still underestimated, but more realistic. This approximation on the initial nil DDN content in organisms is therefore not sufficient to explain the huge difference with observations concerning the DDN proportion in traps. Another source of error lies in the implicit representation of the aggregation process made in this study. It has indeed been considered that from day 10, 10 % of all the model variables are allowed to sink in addition to the detrital particulate compartment.

However, it seems that this leads to an underestimation of UCYN-C sinking. As already mentioned, Bonnet et al. (a) (subm., this issue) showed that the UCYN-C contribution to the particulate C collected in traps on day 17 was up to $22.4 \pm 4.0$ % against 0.25 % for the model. The *in situ* value has been estimated using a value of the intracellular C-content per cell of 22 pgC.cell$^{-1}$ determined according to the measured UCYN-C cell size in the mesocosms and the equations of Verity et al. .

However, the modelled C intracellular content of UCYN-C at day 17 is about 150 times lower (0.13 pgC.cell$^{-1}$). This difference in UCYN-C C-contents is due to the straightforward hypothesis we made in the model which was to consider the UCYN-C diazotrophs as PHYS. Our aim was indeed to use the same model developed for the oligotrophic ocean and particularly the Mediterranean Sea (Eco3M-MED) in every oligotrophic region of the ocean. Moreover, we considered that it was po-

tentially informative to consider that the diazotrophs added in the model were similar in all points to PHYS and PHYL except that there were able to fix $N_2$. In the model, PHYS represents picophytoplankton and the small nanophytoplankton, and its C intracellular content ranges between 0.08 and 0.25 pgC.cell$^{-1}$, which seems to be an underestimated value for UCYN-C. During the VAHINE experiment, large cells of UCYN-C (size about 5.7 $\mu$m (Bonnet et al. (b), subm., this issue)) were

present with a C content estimated at 22 pgC.cell$^{-1}$ (Bonnet et al. (a), subm., this issue). With the latter C content, we established that the mUCYN-C contribution to export would reach 28 %, a result consistent with the $22.4 \pm 4.0$ % estimated by Bonnet et al. (a) (subm., this issue). Finally, the overestimation of UCYN-C abundance by the model also supports the idea that UCYN-C sinking is underestimated by the model. The aggregation process induced by TEP (Berman-Frank et al., subm.,

this issue) or by specific molecules such as extracellular polysaccharides (EPS) (Sohm et al., 2011), which is not explicitly represented in the model, might explain the preferential export of UCYN-C in the mesocosms. Hence, though aggregation is probably overestimated in the mesocosms as compared to natural situations, the contribution of this process seems to be significant in C export. Overall, though the clear underestimation by the model of the UCYN-C sinking and DDN export,

the main conclusions delivered by the model concerning the fate of DDN through the planktonic food web remain unchanged.





### 4.4 Conclusion

The DIP enrichment conducted during the VAHINE mesocosms experiment in the oligotrophic water of the New Caledonia lagoon (SW Pacific Ocean) led to a clear increase in primary, bacterial and

export productions. Two simulations, with and without considering the DIP enrichment, were run. Their comparison allowed to quantify the increase in the main biogeochemical fluxes due to the DIP enrichment. This modeling work was also intended to investigate the fate of the N provided by $N_2$ fixation (i.e. DDN) throughout the planktonic food web. The dynamics of the functional groups provided by the simulation with the DIP enrichment is globally consistent with the measured ones,

especially the development of UCYN-C 10 days after the DIP enrichment. The lag time of 10 days (concomitant with the increase in primary, bacterial and export productions) raises the question of the relevance of the classical methods used to quantify primary and bacterial nutrient limitation, at least in areas where $N_2$ fixation may sustains a large proportion of new PP. This modeling study also allowed to follow the fate of the new N input by $N_2$ fixation (DDN) in the ecosystem. According to

the model, DDN is mainly found in the dissolved pool ($NH_4^+$ and DON) before benefiting the whole planktonic community. At the end of the simulation, 43 %, 33 %, and 15 % are respectively found in non-diazotroph organisms, UCYN-C and DON, respectively. The exported matter collected in the traps at 15 m depth showed that export is essentially due to the sinking of small organisms. Although the measured and simulated C, N, P export are consistent in magnitude, the simulated percentage of

DDN in traps is significantly lower than that of experimental measurements. During the experiment, UCYN-C export was high, probably due to their aggregation in larger particles thanks to the secretion of TEP or EPS which increased their own sinking velocity rather than the sinking velocity of the whole suspended matter as considered in the present model. Directly or indirectly, small diazotrophs significantly contribute to the particulate export through the aggregation process which needs to be

further investigated in future work.



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










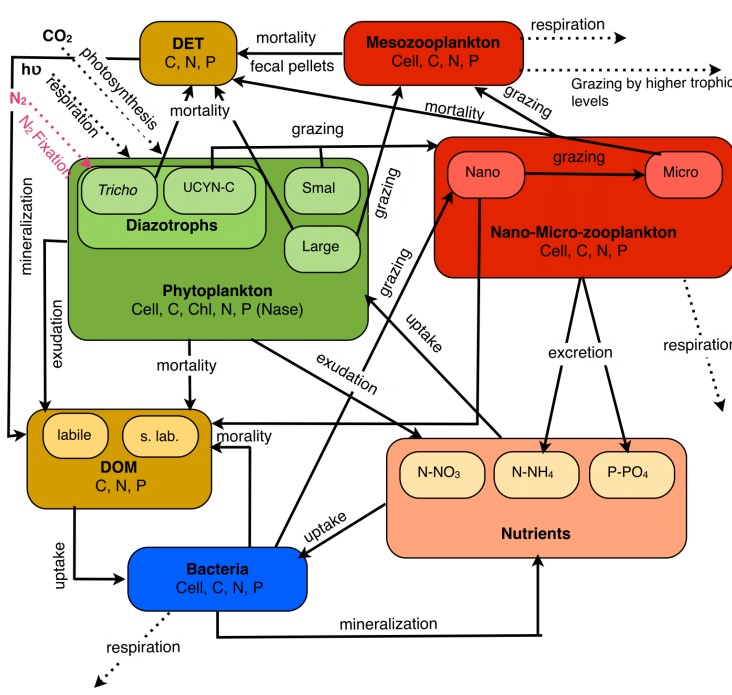

**Figure 1.** Conceptual diagram of the biogeochemical model for the 1D-vertical model used in theVAHINE experiment.





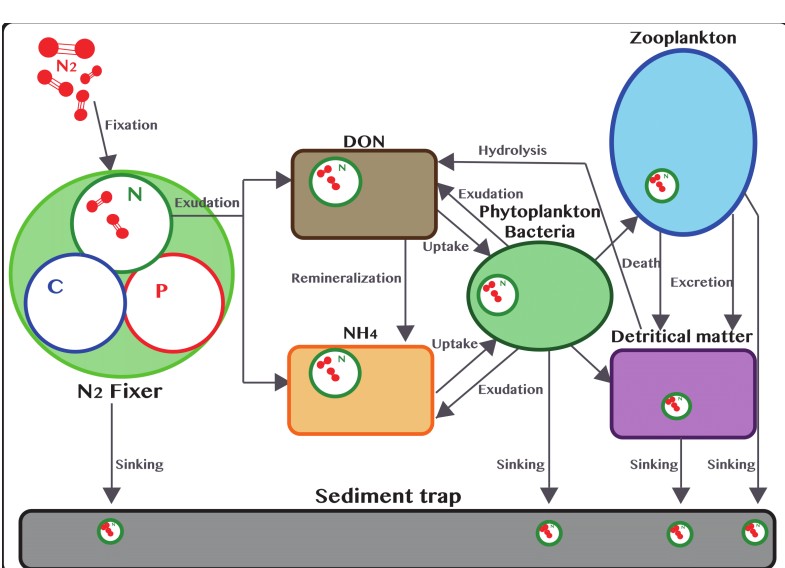

**Figure 2.** Conceptual diagram of the DDN pathway with compartments and processes engaged in the DDN transfer within the food web.





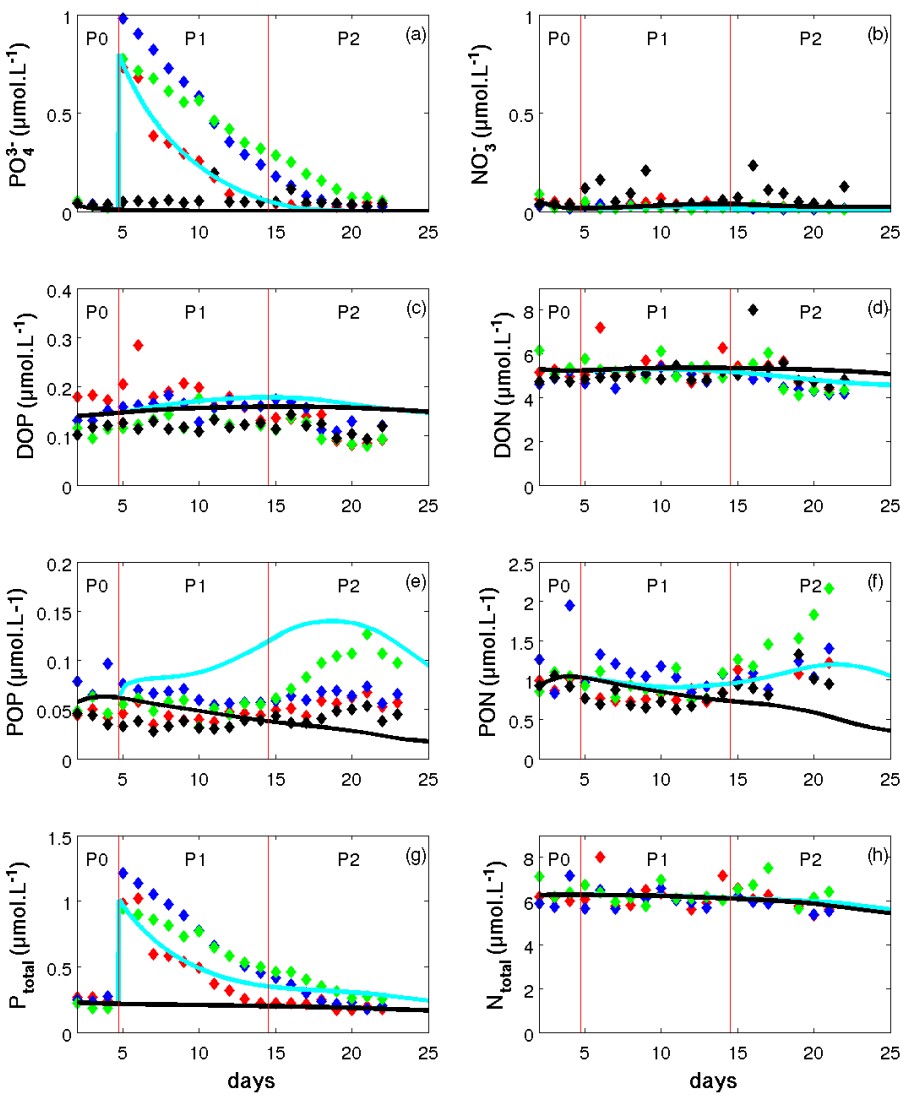

**Figure 3.** Temporal evolution of (a) dissolved inorganic phosphate (DIP), (b) nitrate ($NO_3^-$), (c) dissolved organic phosphate (DOP), (d) dissolved organic nitrogen (DON), (e) particulate organic phosphorus (POP), (f) particulate organic nitrogen (PON), (g) total phosphorus ($P_{total}$) and (h) total nitrogen ($N_{total}$) concentrations ($\mu$mol.L$^{-1}$) in model outputs (solid lines : SIM$^E$ (blue) and SIM$^C$ (black)) averaged on depth superposed to data observations averaged on depth in the three mesocosms (M1 (red), M2 (blue), M3 (green)) and in surrounding waters (black). Red vertical lines distinguish the three periods P0 (before the DIP enrichment), P1 (Diatom-diazotroph associations dominate the diazotrophic community) and P2 (unicellular $N_2$-fixing cyanobacteria (Group C) dominate the diazotrophic community).





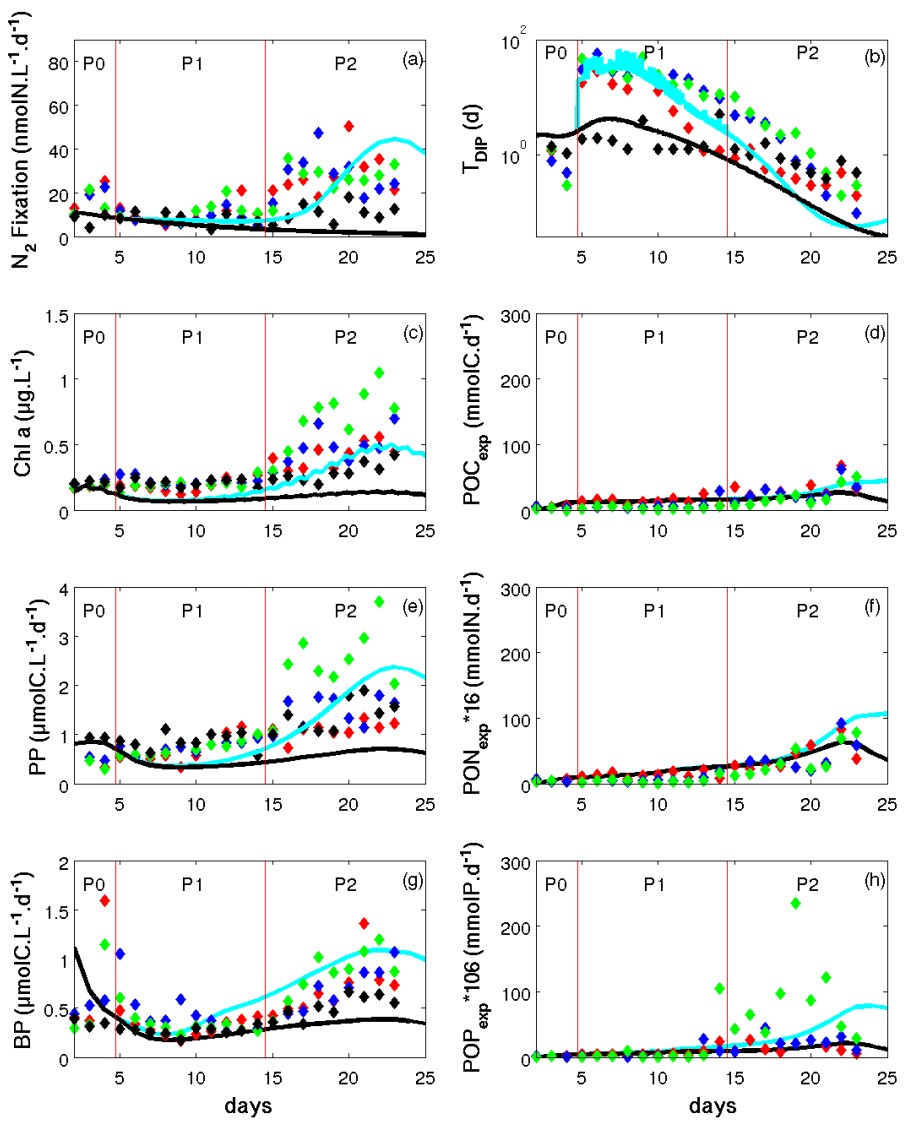

**Figure 4.** Temporal evolution of (a) dinitrogen fixation (N$_2$ fixation) rates (nmolN.L$^{-1}$.d$^{-1}$) , (b) dissolved inorganic phosphate turnover time (T$_{DIP}$, days) , (c) chlorophyll a (Chl a, μg.L$^{-1}$), (d) particulate organic C exported (POC$_{exp}$, dry matter in mmolC ), (e) primary production (PP) rates (μmolC.L$^{-1}$.d$^{-1}$), (f) particulate organic nitrogen exported * 16 (PON$_{exp}$, dry matter in mmolN ), (g) Bacterial production (BP) rates (μmolC.L$^{-1}$.d$^{-1}$) and (h) particulate organic phosphate exported * 106 (POP$_{exp}$, dry matter in mmolN ) in model outputs (solid lines : SIM$^E$ (blue) and SIM$^C$ (black)) averaged on depth superposed to data observations averaged on depth in the three mesocosms (M1 (red), M2 (blue), M3 (green)) and in surrounding waters (black). Red vertical lines distinguish the three periods P0 (before the DIP enrichment), P1 (Diatom-diazotroph associations dominate the diazotrophic community) and P2 (unicellular N$_2$-fixing cyanobacteria (Group C) dominate





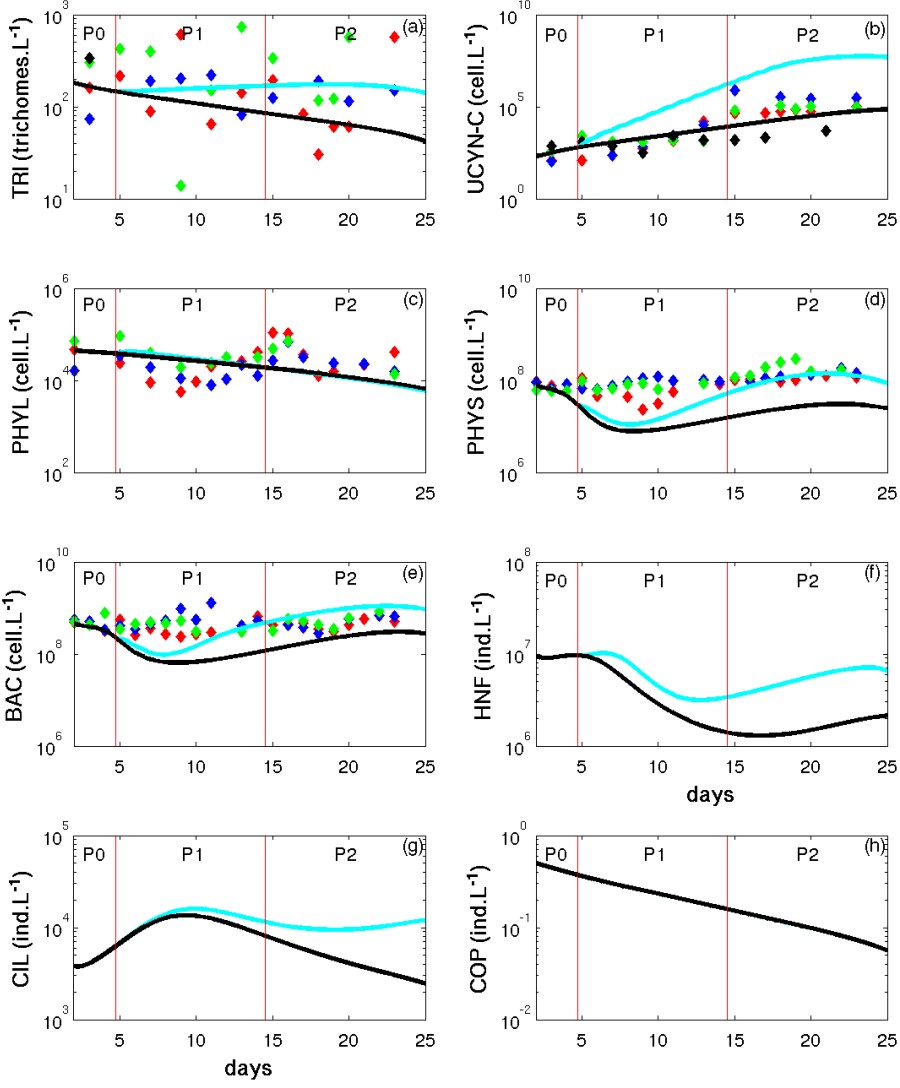

**Figure 5.** Abundances temporal evolution of (a) *Trichodesmium* (TRI, trichom.L$^{-1}$), (b) unicellular N$_2$-fixing cyanobacteria (UCYN-C, cell.L$^{-1}$), (c) large phytoplankton (PHYL, cell.L$^{-1}$), (d) small phytoplankton(PHYS, cell.L$^{-1}$), (e) heterotrophic bacteria (BAC, cell.L$^{-1}$), (f) hetero-nanoflagellates (HNF, ind.L$^{-1}$), ciliates (CIL, ind.L$^{-1}$) and copepods (COP, ind.L$^{-1}$) in model outputs (solid lines : SIM$^E$ (blue) and SIM$^C$ (black)) averaged on depth superposed to data observations averaged on depth in the three mesocosms (M1 (red), M2 (blue), M3 (green)) and in surrounding waters (black). Red vertical lines distinguish the three periods P0 (before the DIP enrichment), P1 (Diatom-diazotroph associations dominate the diazotrophic community) and P2 (unicellular N$_2$-fixing cyanobacteria (Group C) dominate the diazotrophic community).





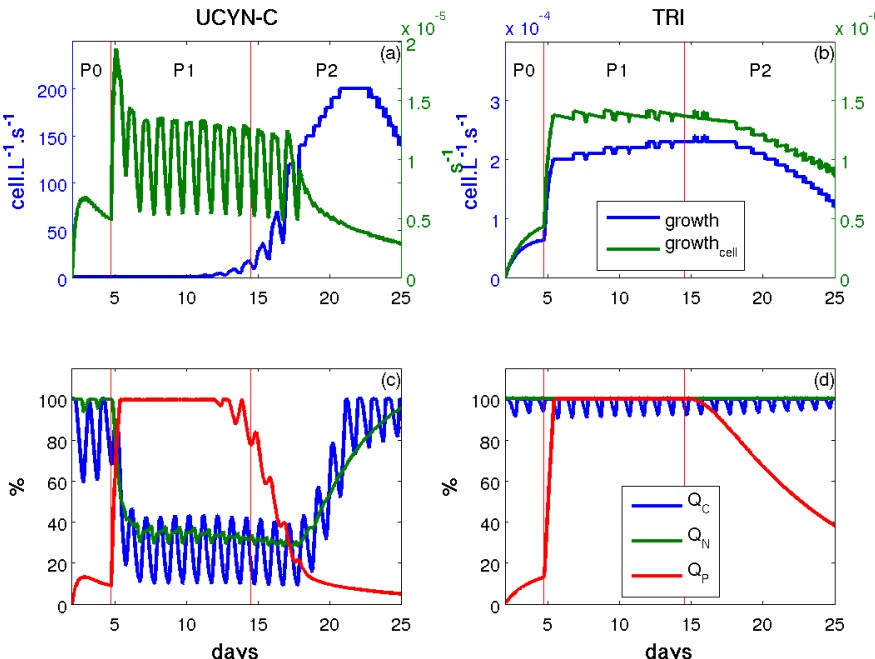

**Figure 6.** Temporal evolution of specific- (green) and population (blue) growth rates function of (a) unicellular $N_2$-fixing cyanobacteria (UCYN-C, cell.s$^{-1}$) and (b) *Trichodesmium* (TRI, trichome.s$^{-1}$) and carbon (C, blue), nitrogen (N, green) and phosphorus (P, red) relative intracellular quota in (c) unicellular $N_2$-fixing cyanobacteria (Group C) (UCYN-C, %) and (d) *Trichodesmium* (TRI, %) in model outputs in SIM$^E$. Red vertical lines distinguish the three periods P0 (before the DIP enrichment), P1 (Diatom-diazotroph associations dominate the diazotrophic community) and P2 (unicellular $N_2$-fixing cyanobacteria (Group C) dominate the diazotrophic community).





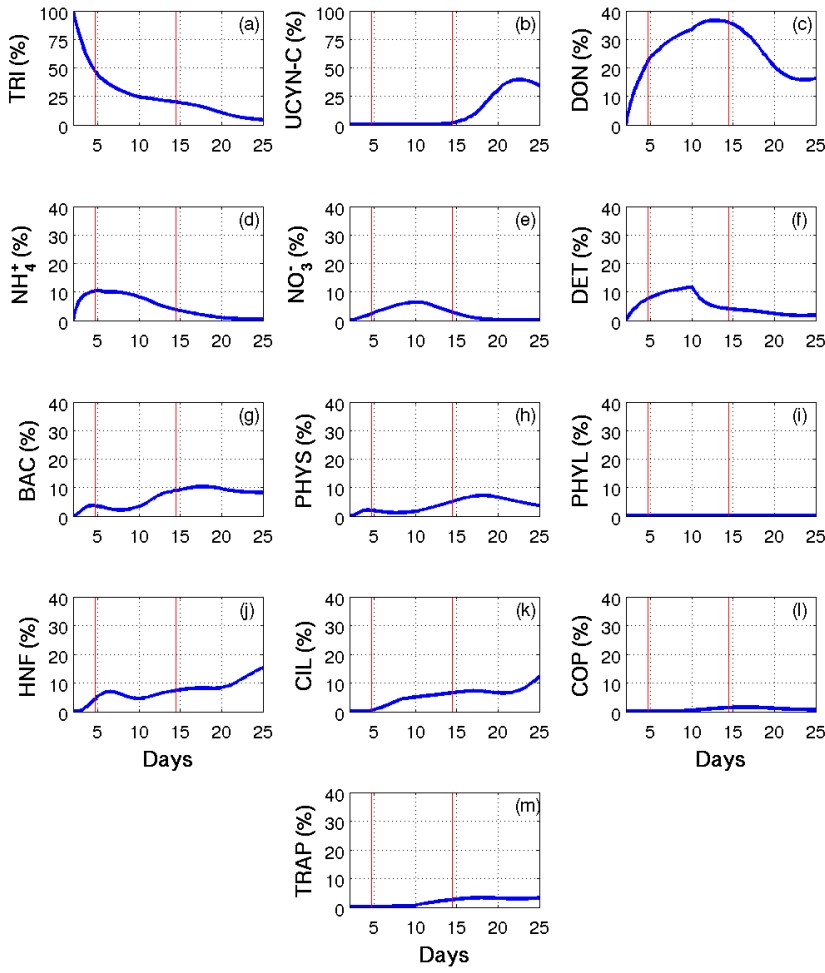

**Figure 7.** Temporal evolution of DDN proportion (%) in (a) *Trichodesmium* (TRI), (b) unicellular N$_2$-fixing cyanobacteria (Group C) (UCYN-C), (c) dissolved organic nitrogen (DON), (d) ammonium (NH$_4^+$), (e) nitrate (NO$_3^-$) (f) detrital nitrogen (DET$_N$), (g) heterotrophic bacteria (BAC), (h) small phytoplankton (PHYS), (i) large phytoplankton (PHYL), (j) hetero-nanoflagellates (HNF), (k) ciliates (CIL), (l) copepods (COP) and (m) in traps (TRAP) in SIM$^E$. Red vertical lines distinguish the three periods P0 (before the DIP enrichment), P1 (Diatom-diazotroph associations dominate the diazotrophic community) and P2 (unicellular N$_2$-fixing cyanobacteria (Group C) dominate the diazotrophic community)



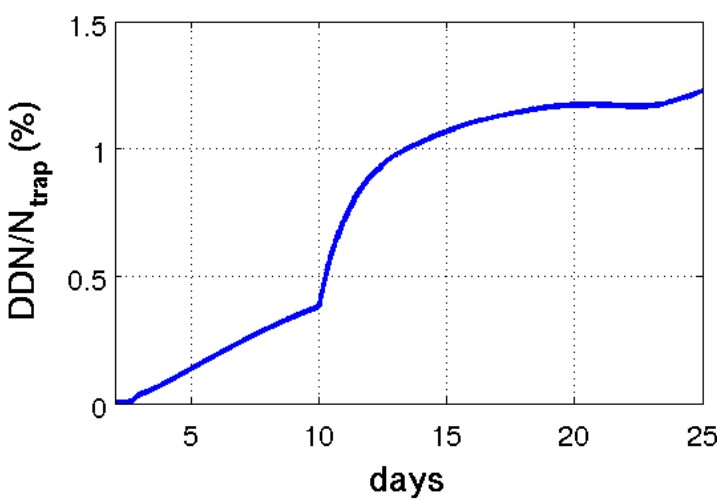

**Figure 8.** Temporal evolution of the nitrogen fixation contribution (%) to particulate matter export in $\text{SIM}^E$.



| State variable | Reference | Value | Unit | State variable | Reference | Value | Unit |
|---|---|---|---|---|---|---|---|
| $BAC_{cell}$ | Data | $4.75\,10^{8}$ | $cell.L^{-1}$ | $HNF_{cell}$ | $\frac{BAC_{cell}}{50}$ | $9.519\,10^{6}$ | $ind.L^{-1}$ |
| $BAC_C$ | $BAC_{cell} \times Q_C\,max$ | 1.152 | $\mu molC.L^{-1}$ | $HNF_C$ | $HNF_{cell} \times Q_C\,max$ | 34.950 | $\mu molC.L^{-1}$ |
| $BAC_N$ | $BAC_{cell} \times Q_N^{moy} \times 0.7$ | 0.107 | $\mu molN.L^{-1}$ | $HNF_N$ | $HNF_{cell} \times Q_N^{moy} \times 0.7$ | 0.326 | $\mu molN.L^{-1}$ |
| $BAC_P$ | $BAC_{cell} \times Q_P^{min}$ | 0.007 | $\mu molP.L^{-1}$ | $HNF_P$ | $HNf_{cell} \times Q_P^{min}$ | 0.023 | $molP.L^{-1}$ |
| $CIL_{cell}$ | $\frac{HNF_{cell}}{2500}$ | 3808 | $ind.L^{-1}$ | $MZOO_{cell}$ | Adapted data | 0.5 | $ind.L^{-1}$ |
| $CIL_C$ | $CIL_{cell} \times Q_C\,max$ | 1.538 | $\mu molC.L^{-1}$ | $MZOO_C$ | $MZOO_{cell} \times Q_C^{max}$ | 0.350 | $\mu molC.L^{-1}$ |
| $CIL_N$ | $CIL_{cell} \times Q_N^{moy} \times 0.7$ | 0.108 | $\mu molN.L^{-1}$ | $MZOO_N$ | $MZOO_{cell} \times Q_N^{moy} \times 0.7$ | 0.042 | $\mu molN.L^{-1}$ |
| $CIL_P$ | $CIL_{cell} \times Q_P^{min}$ | 0.005 | $\mu molP.L^{-1}$ | $MZOO_P$ | $MZOO_{cell} \times Q_P^{min}$ | 0.002 | $\mu molP.L^{-1}$ |
| $PHYL_{cell}$ | Data | $4.48\,10^{4}$ | $cell.L^{-1}$ | $PHYS_{cell}$ | Data | $8.11\,10^{7}$ | $cell.L^{-1}$ |
| $PHYL_C$ | $PHYL_{cell} \times Q_C^{max}$ | 0.306 | $\mu molC.L^{-1}$ | $PHYS_C$ | $PHYS_{cell} \times Q_C^{max}$ | 1.664 | $\mu molC.L^{-1}$ |
| $PHYL_N$ | $PHYL_{cell} \times Q_N^{moy} \times 0.7$ | 0.022 | $\mu molN.L^{-1}$ | $PHYS_N$ | $PHYS_{cell} \times Q_N^{moy} \times 0.7$ | 0.117 | $\mu molN.L^{-1}$ |
| $PHYL_P$ | $PHYL_{cell} \times Q_P^{min}$ | $9.634\,10^{-4}$ | $molP.L^{-1}$ | $PHYS_P$ | $PHYS_{cell} \times Q_P^{min}$ | 0.005 | $\mu molP.L^{-1}$ |
| $PHYL_{Chl}$ | $\frac{PHYL_C}{25}$ | 0.012 | $\mu gChl.L^{-1}$ | $PHYS_{Chl}$ | $\frac{PHYS_C}{12}$ | 0.138 | $\mu gChl.L^{-1}$ |
| $UCYN-C_{cell}$ | Data | 210 | $cell.L^{-1}$ | $TRI_{cell}$ | Data | 180 | $cell.L^{-1}$ |
| $UCYN-C_C$ | $UCYN-C_{cell} \times Q_C^{max}$ | 4.308 | $pmolC.L^{-1}$ | $TRI_C$ | $TRI_{cell} \times Q_C^{max}$ | | $molC.L^{-1}$ |
| $UCYN-C_N$ | $UCYN-C_{cell} \times Q_N^{max}$ | 0.650 | $pmolN.L^{-1}$ | $TRI_N$ | $TRI_{cell} \times Q_N^{max}$ | | $molN.L^{-1}$ |
| $UCYN-C_P$ | $UCYN-C_{cell} \times Q_P^{min}$ | 0.013 | $pmolP.L^{-1}$ | $TRI_P$ | $TRI_{cell} \times Q_P^{min}$ | | $molP.L^{-1}$ |
| $UCYN-C_{Nase}$ | $\frac{TRI_{Nase}}{33300}$ | $1.9\,10^{20}$ | $molN.cell^{-1}.s^{-1}$ | $TRO_{Nase}$ | Rabouille et al. 2006 | $7.5\,10^{16}$ | $molN.trich^{-1}.s^{-1}$ |
| $UCYN-C_{Chl}$ | $\frac{UCYN-C_C}{12}$ | 0.359 | $pgChl.L^{-1}$ | $TRI_{Chl}$ | $\frac{TRI_C}{25}$ | | $gChl.L^{-1}$ |
| labile DOC | Data | 0.25 | $\mu molC.L^{-1}$ | $POC_{Det}$ | $POC_{Tot} - POC_{Living}$ | | $\mu molC.L^{-1}$ |
| semi labile DOC | labile DOC $\times 19$ | 4.75 | $\mu molC.L^{-1}$ | $PON_{Det}$ | $PON_{Tot} - PON_{Living}$ | | $\mu molN.L^{-1}$ |
| labile DON | Data | 1.0 | $\mu molN.L^{-1}$ | $POP_{Det}$ | $POP_{Tot} - POP_{Living}$ | | $\mu molP.L^{-1}$ |
| labile DOP | Data | 0.0132 | $\mu molP.L^{-1}$ | | | | |
| $NO_3^-$ | Data | 53 | $nmol.L^{-1}$ | | | | |
| $NH_4$ | Data | 36 | $nmol.L^{-1}$ | | | | |
| $PO_4$ | Data | 30 | $nmol.L^{-1}$ | | | | |

**Table 1.** Initial conditions for the biogeochemical model



| Parameter | Definition | TRI value | UCYN-C value | unit |
|---|---|---|---|---|
| $Q_C^{min}$ | minimum quota of C | $2.28\ 10^{-10}$ | $6.84\ 10^{-15}$ | $molC.Cell^{-1}$ |
| $Q_C^{max}$ | maximum quota of C | $6.84\ 10^{-15}$ | $2.05\ 10^{-14}$ | $molC.Cell^{-1}$ |
| $Q_N^{min}$ | minimum quota of N | $3.44\ 10^{-11}$ | $1.03\ 10^{-15}$ | $molN.Cell^{-1}$ |
| $Q_N^{max}$ | maximum quota of N | $1.03\ 10^{-10}$ | $3.09\ 10^{-15}$ | $molN.Cell^{-1}$ |
| $Q_P^{min}$ | minimum quota of P | $3.44\ 10^{-11}$ | $1.03\ 10^{-15}$ | $molN.Cell^{-1}$ |
| $Q_P^{max}$ | maximum quota of P | $1.03\ 10^{-10}$ | $3.09\ 10^{-15}$ | $molN.Cell^{-1}$ |
| | | | | |
| $Q_{CN}^{min}$ | minimum C:N ratio | 5.0 | 5.0 | $molC.molN^{-1}$ |
| $Q_{CN}^{max}$ | maximum C:N ratio | 19.8 | 19.8 | $molC.molN^{-1}$ |
| $Q_{CP}^{min}$ | minimum C:P ratio | 35.33 | 35.33 | $molC.molP^{-1}$ |
| $Q_{CP}^{max}$ | maximum C:P ratio | 318.0 | 318.0 | $molC.molP^{-1}$ |
| | | | | |
| $\mu_{max}$ | maximum growth rate | $2.08\ 10^{-6}$ | $3.2\ 10^{-5}$ | $s^{-1}$ |
| $k_m$ | specific natural mortality rate | $1.16\ 10^{-6}$ | $1.16\ 10^{-6}$ | $s^{-1}$ |
| | | | | |
| $K_{NO^3}$ | Half-saturation constant for $NO^3$ | $1.85\ 10^{-6}$ | $7.6\ 10^{-6}$ | $mol.L^{-1}$ |
| $V_{NO^3}^{max}$ | Maximum uptake rate for $NO^3$ | $3.16\ 10^{-15}$ | $9.91\ 10^{-20}$ | $mol.cell^{-1}.s^{-1}$ |
| $K_{NH^4}$ | Half-saturation constant for $NH^4$ | $7.0\ 10^{-6}$ | $1.69\ 10^{-6}$ | $mol.L^{-1}$ |
| $V_{NH^4}^{max}$ | Maximum uptake rate for $NH^4$ | $3.16\ 10^{-15}$ | $9.91\ 10^{-20}$ | $mol.cell^{-1}.s^{-1}$ |
| $K_{PO^4}$ | Half-saturation constant for $PO^4$ | $1.4\ 10^{-6}$ | $2.62\ 10^{-7}$ | $mol.L^{-1}$ |
| $V_{PO^4}^{max}$ | Maximum uptake rate for $PO^4$ | $1.98\ 10^{-16}$ | $6.19\ 10^{-21}$ | $mol.cell^{-1}.s^{-1}$ |
| $K_{DON}$ | Half-saturation constant for $DON$ | $4.32\ 10^{-5}$ | $1.05\ 10^{-5}$ | $mol.L^{-1}$ |
| $V_{DON}^{max}$ | Maximum uptake rate for $DON$ | $3.16\ 10^{-15}$ | $9.91\ 10^{-20}$ | $mol.cell^{-1}.s^{-1}$ |
| $K_{DOP}$ | Half-saturation constant for $DOP$ | $3.4\ 10^{-6}$ | $6.57\ 10^{-7}$ | $mol.L^{-1}$ |
| $V_{DOP}^{max}$ | Maximum uptake rate for $DOP$ | $3.16\ 10^{-15}$ | $6.19\ 10^{-21}$ | $mol.cell^{-1}.s^{-1}$ |
| | | | | |
| $Nase_{prod}^{max}$ | Maximum rate of increase of nitrogenase activity | $1.17\ 10^{-21}$ | $3.51\ 10^{-26}$ | $mol.cell^{-1}.s^{-2}$ |
| $Nase_{decr}^{max}$ | Maximum rate of decay of nitrogenase activity | $9.36\ 10^{-22}$ | $2.83\ 10^{-26}$ | $mol.cell^{-1}.s^{-2}$ |
| $K_{Nase}$ | Coefficient of nitrogenase degradation | $9.44\ 10^{-16}$ | $1.92\ 10^{-20}$ | $mol.cell^{-1}.s^{-1}$ |
| $COST_{DIAZO}$ | Respiration cost for nitrogen fixation | 1.5 | 1.5 | $mol.mol^{-1}$ |
| $EXUD_{DON}$ | Exsudation part of N$_2$ fixed towards DON | 0.5 | 0.5 | |
| $EXUD_{NH_4}$ | Exsudation part of N$_2$ fixed towards $NH_4$ | 0.5 | 0.5 | |

**Table 2.** Parameters added for the diazotroph organisms