# Peer review of "Biogeochemical fluxes and fate of diazotroph derived nitrogen in the food web after a phosphate enrichment: Modeling of the VAHINE mesocosms experiment"

_Biogeosciences, 2015_

## Referee Comment (RC1) · Anonymous Referee #1 · 25 Feb 2016

This manuscript describes modeling of a mesocosm experiment studying the response, in particular in diazotrophs, to phosphate addition.

If this interpretation of the intention with the experiment is correct, I am worried with the experimental design. There seems to be no control bags(?) It then seems to me that one cannot formally know whether the responses are due to the PO4-enrichment or due to the enclosure in bags? This seems to me to weaken the authors' case on time scales and bioassays. I totally agree that there are important time scale issues to consider when doing bioassays, but I am not even sure that there is any single "correct"

answer, probably there can be different answers for different time-scales, including time scales longer than the one studied here. These worries may seem to be be of more concern for the reports on the experimental data than for this modeling report. It does, however, mean that the model is challenged only with one experimental situation. The work is therefor probably not very strong as a validation of the model. The model gives, however, a background that serves well to structure the discussion of mechanisms involved.

The interactions between P and N, the role of flexible stoichiometry and the difference in grazing pressure on different phytoplankton groups makes an interesting subject for model studies and the model presented here is therefore an interesting contribution.

---

## Referee Comment (RC2) · Anonymous Referee #2 · 12 Mar 2016

This manuscript by Gimenez et al., is a modelling study based on experimental data from the VAHINE mesocosm study where dissolved inorganic phosphate was added to support the growth of diazotrophic (N2-fixing) organisms. The particular focus of this study was to track nitrogen fixed by diazotrophs through the food web over longer time periods that could not be studied through experimental work, in addition to quantifying the flux of carbon, nitrogen and phosphorus in the mesocosm system.

Gimenez et al. used a biogeochemical mechanistic model based on the Eco3M-MED for the Mediterranean Sea where N2-fixation was included as a function of enzyme

activity and diazotroph abundances. Using the results of the model, this study reports that diazotroph-derived nitrogen was initially released to the dissolved organic nitrogen (DON) and ammonium (NH4+) pools. Then it was assimilated into the plankton biomass with the majority (43%) in non-diazotrophic plankton after 25 days in the simulation. It is pleasing to see work on incorporating diazotrophy into models, particularly when supported by in depth information from experimental work such as the VAHINE study. Furthermore, phosphate enrichment enhanced N2-fixation, primary production and the export of carbon by 201, 208 and 87%, respectively compared to the non-enriched simulation. However this enhancement effect from phosphate enrichment had a lag of around 10 days, hence the authors highlight this long time period compared to common methods used to determine nutrient limitation which are usually on much shorter time scales. This is a result that I feel will likely be of interest to others working on nutrient limitation and dynamics in aquatic ecosystems. Thus, I recommend publication of the manuscript once issues detailed below have been addressed.

**GENERAL COMMENTS:**

In this mesocosm study, there was no control mesocosm where no dissolved inorganic phosphorus (DIP) added (Bonnet et al., in review). This is also described in the Methods section (P3, lines 88 – 91) with the non-enriched simulation is "considered as a proxy of the planktonic dynamics outside the mesocoms in lagoon waters" (P9, lines 251 – 252). However from previous largescale mesocosm experiments, there have been considerable differences between the control mesocosms and the surrounding waters, primarily due to entrainment of different water masses, which cannot occur within the closed mesocosm system (see for example recent ocean acidification studies as Special Issues also in Biogeosciences, http://www.biogeosciences.net/special\_issue204.html, http://www.biogeosciences.net/special\_issue120.html). Hence while the different phosphate concentrations are acknowledged in the Discussion (P14, lines 424 - 426), potential phosphate inputs into the sampled lagoon waters is not quantified and the jus-

**BGD**
tification for the use of the non P-enriched simulation as a proxy for the lagoon waters is currently weak.

Despite high variability between the mesocosms in the initial conditions through the DIP addition (see for example Fig. 3(a)), it appears as though the DIP addition simulated is not an average of the amount present in all mesocosms, instead more closely fitting the concentrations in M1 and M3. Indeed, the simulation of DIP concentrations in SIME follows closely to the dynamics in M1. This is a potentially interesting result that is currently not given much attention in the manuscript. In addition, there are also some notable deviations in the temporal evolution of various parameters between the model runs and the experimental data e.g. dissolved inorganic phosphate, UCYN-C abundances, and the abundances of small phytoplankton and bacteria (see also Figs. 3-5). Both of these points could be discussed more in depth in the manuscript.

Language: In general, the manuscript reads well and has a logical structure based on the headings and sub-headings, however it would benefit from proofreading by a native speaker as some phrasings and incorrect grammar hinder readability. Some sections are very long and including paragraphs would make it also easier to read as there is a lot of information to take in in one chunk. Brackets are frequently used, but it would make smoother reading if these were better incorporated into the text. Some of these grammatical errors, and others have been highlighted in the specific comments section below.

There is a mixed use of present and past tenses in the Methods section. I would suggest the authors change this to the past tense and describe what was done. The manuscript would benefit from a thorough check for consistent use of either 'mineral' or 'inorganic' and the use of italics for N2 fixation as well as for correct figure numbering. Care also needs to given to the correct and consistent use of capitals (e.g. Cyanobacteria on P2, line 38 and picoEukaryotes on P6, line 172 should read cyanobacteria and picoeukaryotes, respectively) throughout the manuscript, including in the references to figures, tables and equations. These are also currently do not follow clear guidelines
provided by Biogeosciences for figure and table labelling and referencing.

The term "export" is used in reference to the material collected in the sediment trap at around 15 m deep. This is quite a shallow depth considering the euphotic zone which is a commonly used depth for reporting carbon export data. Hence in this study, it appears to more reflect the "sinking flux" or "potential" export rather than export. I recommend that the use of the term "export" be reconsidered to see if this accurately reflects what was measured during study.

Units: Units of nmolN.m-2.d-1 are used and in some sections scientific notation of exponents is incorrect or is not complete (e.g. P12, lines 350 – 356 "5.108 cell.L-1"). The incomplete notation may be a formatting issue that occurred during the proof-reading process. Nonetheless, I would advise that SI units are used and suggest that all data in scientific notation is checked that it is correctly printed as this was distracting when reading the manuscript.

SPECIFIC COMMENTS:

P1, line 9: It is unclear what kind of P was added during this study. It would be clearer if "P" was changed to "inorganic P" in the abstract that to indicate inorganic P was added to the mesocosms.

P1, line 13: It is unclear what is meant by population scale here. Please specify.

P2, line 2: "Pacific ocean" should read "....Pacific Ocean...".

P2, line 41: The bracket before UCYN-A should be a comma i.e. "...Group B, UCYN-A ...".

P3, line 59: It isn't clear what is meant by affected in "...a water mass affected by diazotroph development...". Please rephrase.

P3, line 69: The 15 in "d15N" should be as a superscript.

P3, line 75: A bracket is missing at the end of "... Group C (UCYN-C)".

BGD
P4, line 93: Please specify if "the sediment traps were collected" or instead "the material from the sediment traps were collected".

P4, lines 112 - 115: Here the sinking velocity was set as constant as 0.7 m/day for the first 10 days but increased according to a polynomial function to up to 10 m/day. Why was this function used and why would the sinking velocity of the particles suddenly increase after 10 days? This appears to have a marked and sudden influence on the model output eg. Fig. 7(c), (f) and (m), that does not seem to fit with the experimental data in Fig. 4.

P5, line 130: Language error - should read 'When the total N and P pools (Ntotal and Ptotal) were calculated from the model outputs...'.

P5, line 153: Is the zooplankton abundance and C, N, P data from this study?

P5, line 143: Language error – "autotroph phytoplankton" should read "autotrophic phytoplankton".

P6, lines 167-170: The labile DON fraction is defined as the "...quantity consumed during the experiment in the mesocosms which was estimated at 1 umol.L-1". How does this definition fit with the probable production of DON during the study period? Could this labile fraction be underestimated?

P8, line 222: The number of cells in a trichome for Trichodesmium sp. is variable rather than a consistent number between trichomes. Hence here, I would suggest using the word "assuming" rather than "considering" here.

P9, line 255: It is unclear what is meant by "diversity parameters". Is this referring to the composition of the plankton community present? Please clarify.

P9, line 257 - 261: The distinction between P0, P1 and P2 is clearly defined in the caption of Fig. 3 but not in the body text. It would be helpful to also have this in the text here as well.

BGD
P9, line 273: Should "nmNH4+" instead read "mNH4+"? Is this correct or is this a typing error. Here "remains" should also read "remained".

P10, line 278: The increase in mDOP in SIME described in the body text is very difficult to distinguish from Fig. 3(c). What was the magnitude of this increase?

P12, lines 370 – 371: Are the growth rates reported in per second (s-1) as described here in the text? Indeed, these seem very high, with reported rates of over 200 cells L-1 s-1 in Fig. 6. Is this correct? Or is 'x 10-4' missing from the y-axis? Additionally, it seems that most of the model outputs are quite smooth with minimal variation on a time scale shorter than one day, apart from the turnover time of DIP in the SIME simulation (Fig. 4 (b)). What is the temporal resolution of the model? Is this consistent across all variables included in the model?

P13, lines 387 – 390: According to Fig. 5(b), UCYN-C was present throughout the study period. Why is the UCYN-C proportion of DDN zero at the beginning of the study period, whereas TRICHO starts with 100%?

P14, lines 444-445: "After benefitting diazotrophs, the DDN inputs benefited to nondiazotrophic organisms." – How does the DDN benefit diazotrophs? This could be more clearly and explicitly phrased.

P14, line 447: The reference to Figure 5(e) appears to be for 4(e) here.

P14, line 452: Here, "synthetized" should read "synthesized".

P15, lines 454 – 456: "As the model does not represent the diatom-diazotroph associations (DDAs), which were the most abundant diazotrophs in the mesocosms during P1 (Turk-Kubo et al., subm., this issue), the modeled export is probably underestimated during P1." From looking at Figure 4 (d), (f), and (h), the modelled data appears to have good agreement with the raw data so I am not sure why the authors suggest that the modelled export is probably underestimated.

P15, line 484: The reference to Figure 5(e) appears to be for 4(e) here.
P15, lines 486-487: From Figure 5(b), it appears as though UCYN-C abundances were sampled on day 15, not day 16 as written here.

P15, lines 488-489: This question: "Which factor may explain the 10 days delay between the DIP enrichment and the large UCYN-C development?" seems like it should be a sub-heading rather than in the body text.

P17, lines 531-532: Is the study by Van Wambeke et al., subm. From this mesocosm study? If yes, "... confirming previous studies ... " does not accurately reflect this.

P19, line 614: The citation "Verity et al." is missing the year of publication.

P19, lines 627 – 629: "Finally, the overestimation of UCYN-C abundance by the model also supports the idea that UCYN-C sinking is underestimated by the model". Could underestimated grazing rates also explain this overestimation of UCYN-C abundances in the model?

P20, line 650: This observation of the majority of DDN in the DON and NH4+ pools from the model output is in good agreement with observations from experimental work which is cited in the introduction (P2, line 49). These observations would also fit nicely to the conclusions drawn from this modelling study but are not cited in the context of the discussion here.

Figure 1: Does the model include mortality of all diazotrophs or just Trichodesmium sp.? Is the arrow from N2 to DOM via mineralisation included in the model? If yes, is this mediated by diazotrophs or is this a separate process as indicated in this figure? How is the uptake of DOP by diazotrophs incorporated in the model? Typing errors: "morality" should be "mortality" and "Smal" should be "Small".

Figure 3: The y-axis label of PO43- in Figure 3 is not consistent with the figure caption which abbreviates phosphate to DIP.

Table 1: How was the living fraction of POM (i.e. "POC-living") calculated?

BGD
Table 2: The superscripts on NO3, NH4 and PO4 should instead be subscripts in both the "Parameter" and "Definition" columns. "Exsudation" should also read "Exudation". I would recommend adding "cell" to the "minimum quota of ..." to read "minimum cell quota of ..."

**REFERENCES:**

Bonnet, S., Moutin, T., Rodier, M., Grisoni, J. M., Louis, F., Folcher, E., Bourgeois, B., Boré, J. M., and Renaud, A.: Introduction to the project VAHINE: VAriability of vertical and tropHIc transfer of diazotroph derived N in the south wEst Pacific, Biogeosciences Discuss., doi:10.5194/bg-2015-615, in review, 2016.

---

## Referee Comment (RC3) · Anonymous Referee #3 · 17 Mar 2016

The study of Gimenez et al. aims to model the change in biogeochemical fluxes and the fate of diazotroph-derived nitrogen (DDN) throughout the planktonic food web of the South West Pacific Ocean during a phosphate-enrichment experiment. The model results are compared with and help understand direct observations of the diazotroph role from the VAHINE project where phosphate-enrichment experiments have been carried out in 3 mesocosms near New Caledonia. The model is a 1D vertical box model which accounts for the growth of 2 main types of nitrogen fixers present in this area (Trichodesmium and UCYN-C). The model has one of the most realistic model representation of nitrogen fixation including prognostic equations for enzyme nitrogenase,

cell quota and a distinction between different temporal diazotrophs ability to protect nitrogenase against O2 damage. Gimenez study presents four main results: (1) diazotrophs (mostly under the form of UCYN) responded with a 10-day delay to the PO4 enrichment, which suggests that traditional short-term nutrient-addition experiments are not long enough to inform on nitrogen fixing dependant community; (2) phosphate addition resulted in a increase in nitrogen fixation which then increased primary production and bacterial production; (3) DDN is mainly found in the dissolved pool (NH4+ and DON) before benefiting the whole planktonic community. After 25 days, 43 %, 33 %, and 15 % are found in non-diazotroph organisms, UCYN-C and DON, respectively; and, finally (4) there is a strong impact of UCYN increase on export production probably via the aggregation of small particles due to TED production, effect on the rest of the food web

This work is the first modelling study that look at the fate of diazotroph-derived nitrogen throught the food web, which is an important aspect of the nitrogen cycle as nitrogen fixation is often recognised as an important source of nitrogen to the ocean, but is rarely quantified and looked through in details. This modelling work is then novel and key for the ocean biogeochemical community. In particular, it focuses on the South West Pacific Ocean, which is one of the highest nitrogen fixing area of the global ocean, and has direct observations to compare the model results with. I thus strongly recommend publication of this work. I have however some reservations related to the 10-day delay model result and how concise the manuscript is currently written.

General comments:

*10-day delay: This is an important result suggesting that there is a 10-day lag between the phosphate enrichment and the increase in diazotrophs. First, it is often confused in the text the response between nitrogen fixation and diazotroph concentration, which needs to be corrected. It is not diazotrophs but nitrogen fixation that responds with a delay. Figure 5 shows that Trichosdesmium and UCYNC increase concentration right after the phosphate addition (when looking at model results). In addition if the authors

base their results on nitrogen fixation rate, there is the issue that the model does not match the observations for nitrogen fixation: observed nitrogen fixation rises 5 days after the enrichment (day 10, based on Figure 5), whereas modelled nitrogen fixation rises 10 days after. To conclude that there is a 10-day delay is quite an uncertain result, which needs to be clearly mentioned and discussed. Finally, the text does not describe accurately the results in the model and data: "Since we observed the increase in PP and BP only after 10 days in both experimental and simulation results" (line 537). This is not what Figure 4 shows and should be amend accordingly as well as tone down the conclusion "we may conclude that 10 days are necessary for the newly fixed N by diazotrophs to sustain the observed high production rates, and to see an effective change in the planktonic populations".

*UCYN as the dominant nitrogen fixers: It is not clear from the presented model results (mainly figure 5) that UCYN-C is the dominant nitrogen fixers. I agree that the increase is stronger for UCYN than Trichodesmium, but there isn't enough presented evidence that UCYN explains most of the nitrogen fixation increase (especially considering that Trichodesmium have a larger volume than UCYN). Can you provide more evidence to validate this important point?

*Length of the manuscript: While the structure of the manuscript is good (logical, well-organised), the sections are overall too long with too many not necessary relevant details. This makes the manuscript difficult to follow. I would encourage the authors to shorten the paper as much as possible while staying precise and concise.

Specific comments:

- The 1D model has 14 levels of 1 m each, so represents only the first 14 m of the water column. This seems quite shallow as most places the mixed layer is 50-200 m deep. Can you justify this choice?

- There are too many acronyms making the reading quite difficult. Can you use the full name at the start of a new section at least on the key messages?

- How do you know that phosphate is not used directly by other organisms and that the increase in PP, Chla is due to DDN?

- Part 4.1: Mismatch between the description of model results and observations

- Figure 6: Units overlapping

- Why if the population growth rate start only at day 5 but UCYN-C are growing earlier?

- Why not include DDA in the model since it looks like they play a big part in the first part of the experiment?

- Can you justify the location of the study in the abstract?

- Line 34: Tone down the "only few" as there are quite a few models now that incorporate state variables for nitrogen fixers. Add some more references.

- Line 43: also DDA were modelled in Monteiro et al. (2010, 2011)

- Line 100: Can you say which boundary conditions were implemented for Eco3M?

- Section 3.1 is a good description of the results, but it needs to be specified more clearly if they are model or data results.

- Line 370: Can you define more what the difference is between specific growth and population growth as important later on to describe the model results?

- Line 378: Define fgrowthTRItrich and fgrowthTri

- Figure 7: Is that the correct initial condition in the sense that there are more TRI than UCYNC? Why TRI never come back even after the PO4 enrichment. Do observations show higher TRI concentration than UCYN?

- Line 423: It is not obvious from the observations (Figure 4a, black diamonds) that there is a decrease in nitrogen fixation. Can you please explain/amend?

- Line 439: "the DIP enrichment resulted in an increase in the abundances of all planktonic groups except PHYL and COP". This is not what the observations show for PHYL.

- Looking at Figure 4, the system looks like it is loosing DIP during the standard model run, a trend not present in the observations. Have you thought about this feature and do you think it might impact your model results (for ex, in relation to TRI, PHYL and COP)?

---

## Author Comment (AC1) · 21 Apr 2016

The three answers to the referees are gathered in the PDF file attached to a better reading.

Please also note the supplement to this comment:
http://www.biogeosciences-discuss.net/bg-2015-611/bg-2015-611-AC1-supplement.pdf

---

## Author Comment (AC2) · 21 Apr 2016

Dear referee,

We thank you very much for the time you have devoted to reviewing our paper and for the constructive comments and suggestions which have helped us to improve the manuscript. We have addressed the concerns in a point by point response below (comments are copied with our replies below).

Best Regards,

Audrey Gimenez

**Anonymous Referee #1**

**Referee #1:**

**This manuscript describes modeling of a mesocosm experiment studying the response, in particular in diazotrophs, to phosphate addition.**

**If this interpretation of the intention with the experiment is correct, I am worried with the experimental design. There seems to be no control bags(?) It then seems to me that one cannot formally know whether the responses are due to the PO4-enrichment or due to the enclosure in bags? This seems to me to weaken the authors' case on time scales and bioassays. I totally agree that there are important time scale issues to consider when doing bioassays, but I am not even sure that there is any single "correct" answer, probably there can be different answers for different time-scales, including time scales longer than the one studied here. These worries may seem to be be of more concern for the reports on the experimental data than for this modeling report. It does, however, mean that the model is challenged only with one experimental situation. The work is therefor probably not very strong as a validation of the model. The model gives, however, a background that serves well to structure the discussion of mechanisms involved.**

**The interactions between P and N, the role of flexible stoichiometry and the difference in grazing pressure on different phytoplankton groups makes an interesting subject for model studies and the model presented here is therefore an interesting contribution.**

Answer :

We fully agree that the lack of a mesocosm without DIP enrichment during the experiment is a weak point of this study and we would have liked to include this control if it had been possible to manage more than 3 mesocosms. The main objective of the VAHINE project was to study the input and fate of N by N2 fixation in the mesocosm, and the route of this new nitrogen throughout the entire ecosystem. DIP addition was undertaken to prevent any DIP limitation and to stimulate $N_2$ fixation in order to be better able to follow its fate in the mesocosms.

A modelling approach was carried out in order to help answering these questions and to provide additional information on biological processes. Even if there were no mesocosms without DIP addition, the model enabled us to show the impact of the DIP enrichment on the ecosystem by comparing the enriched and non-enriched simulations, as described in the paper. Concerning the comparison of the simulation without the DIP enrichment with the surrounding waters, we however considered that data from the surrounding waters provided the opportunity to further validate the model under dramatically different nutrient conditions. This compartment proved to be relevant by the fact that the model outputs of the non P-enriched simulation were close to the data collected in the surrounding waters (aware nonetheless that the model did not take into account the lagoon physical processes).

In conclusion, the following sentence has been added in the manuscript to provide better justification of this choice, as mentioned in an answer for referee #2:

"$SIM^C$ outputs have been compared to the data from the surrounding waters where DIP concentration remained very low and constant throughout the experiment. Since mesocosms do not include hydronamical processes, this is merely an approximation. However this comparision provided the opportunity to further validate the model under dramatically different nutrient conditions". (In place of lines 250-252).

This study does not claim to provide a single answer concerning the relevant time scale for enrichment experiments. It rather higlights the strong link between the time scale of the biological response and the biological scale (from cell to population), and aims to question the classical time-scale of enrichment experiments (generally less than two days) since this two days are probably not always sufficient to see a response at population level, at least in our experiment.

**Anonymous Referee #2**

**GENERAL COMMENTS:**
**Referee #2:**

**In this mesocosm study, there was no control mesocosm where no dissolved inorganic phosphorus (DIP) added (Bonnet et al., in review). This is also described in the Methods section (P3, lines 88 – 91) with the non-enriched simulation is "considered as a proxy of the planktonic dynamics outside the mesocoms in lagoon waters" (P9, lines 251 – 252). However from previous largescale mesocosm experiments, there have been considerable differences between the control mesocosms and the surrounding waters, primarily due to entrainment of different water masses, which cannot occur within the closed mesocosm system (see for example recent ocean acidification studies as Special Issues also in Biogeosciences, http://www.biogeosciences.net/special_issue204.html, http://www.biogeosciences.net/special_issue120.html). Hence while the different phosphate concentrations are acknowledged in the Discussion (P14, lines 424 - 426), potential phosphate inputs into the sampled lagoon waters is not quantified and the justification for the use of the non P-enriched simulation as a proxy for the lagoon waters is currently weak.**

Answer :

We acknowledge that the lack of a mesocosm without DIP enrichment during the experiment is a weak point in our study. The comparison of this mesocosm without DIP enrichment with the non P-enriched simulation would have been made if it had been possible to manage more than 3 mesocosms. Aware of this limitation, we however considered that data from the surrounding waters provided the opportunity to further validate the model under dramatically different nutrient conditions. We feel that the fact that the model outputs of the non P-enriched simulation were close to the data collected in the surrounding waters justifies this choice (aware nonetheless that the model did not take into account the lagoon physical processes).

In conclusion, the following sentence has been added in the manuscript to provide better justification of this choice :

"SIM$^C$ outputs have been compared to the data from the surrounding waters where DIP concentration remained very low and constant throughout the experiment. Since mesocosms do not include hydronamical processes, this is merely an approximation. However this comparision provided the opportunity to further validate the model under dramatically different nutrient conditions". in place of lines 250-252.

**Referee #2:**

**Despite high variability between the mesocosms in the initial conditions through the DIP addition (see for example Fig. 3(a)), it appears as though the DIP addition simulated is not an average of the amount present in all mesocosms, instead more closely fitting the concentrations in M1 and M3. Indeed, the simulation of DIP concentrations in SIME follows**

**closely to the dynamics in M1. This is a potentially interesting result that is currently not given much attention in the manuscript. In addition, there are also some notable deviations in the temporal evolution of various parameters between the model runs and the experimental data e.g. dissolved inorganic phosphate, UCYN-C abundances, and the abundances of small phytoplankton and bacteria (see also Figs. 3-5). Both of these points could be discussed more in depth in the manuscript.**

Answer :

Indeed, the DIP concentration after the enrichment on day 5 averaged on the 3 mesocosms is actually 0.83 μM. Moreover, considering the variability in the initial conditions collected over the 3 mesocosms, we decided to run a 'mean simulation' that would not be the best fit of experiments but that would rather provide the general trends of the experiment. As the enrichment brought a large amount of DIP (0,8 μM), a difference of 0.03 μM does not really influence the system dynamics, as we can see in the following figure, which compares the DIP concentrations in the simulation presented in the paper with the one provided by a new simulation run with a DIP enrichment of 0.83 μM.

[Figure]

Concerning the deviations between the model outputs and the experimental data for DIP, UCYN-C, PHYS and BAC abundances, as already mentioned, we did not look for the best fit between model outputs and data (for example, we did not tune any parameter) and the biogeochemical model is the same as the one used for the Mediterranean Sea (except the addition of diazotrophs in the present version). We rather aimed at obtaining a tool that would provide realistic tendencies and orders of magnitude. As a consequence, we did not discuss all the differences between the model outputs and data, but only the most significant. The overestimation of the UCYN-C abundances is indeed discussed in the Discussion (4.3). .

**Language:**

**Referee #2:**

**In general, the manuscript reads well and has a logical structure based on the headings and sub-headings, however it would benefit from proofreading by a native speaker as some phrasings and incorrect grammar hinder readability. Some sections are very long and including paragraphs would make it also easier to read as there is a lot of information to take in in one chunk. Brackets are frequently used, but it would make smoother reading if these were better incorporated into the text. Some of these grammatical errors, and others have been highlighted in the specific comments section below.**

Answer :

We thank the referee very much for these comments about the form of the manuscript. These suggestions have all been taken into account in the revised manuscript.

**Referee #2:**

**There is a mixed use of present and past tenses in the Methods section. I would suggest the authors change this to the past tense and describe what was done.**

Answer :

This suggestion has been taken into account in the revised manuscript. We have now used the past tense everywhere.

**Referee #2:**

**The manuscript would benefit from a thorough check for consistent use of either 'mineral' or 'inorganic' and the use of italics for N2 fixation as well as for correct figure numbering. Care also needs to given to the correct and consistent use of capitals (e.g. Cyanobacteria on P2, line 38 and picoEukaryotes on P6, line 172 should read cyanobacteria and picoeukaryotes, respectively) throughout the manuscript, including in the references to figures, tables and equations. These are also currently do not follow clear guidelines provided by Biogeosciences for figure and table labelling and referencing.**

Answer :

We sincerely apologize for the errors mentioned above. The term 'inorganic' has been chosen to make the reading more consistent in the revised manuscript. In our manuscript, there was no use of italics for the term 'N2 fixation' , we are sorry if this was the case in the downloadable version on the Biogeoscience website. The use of capitals has been checked throughout and corrected in the revised manuscript. Concerning the Biogeosciences guidelines, the manuscript is written with the LateX Biogeoscience package provided on the website.

**Referee #2:**

**The term "export" is used in reference to the material collected in the sediment trap at around 15 m deep. This is quite a shallow depth considering the euphotic zone which is a commonly used depth for reporting carbon export data. Hence in this study, it appears to more reflect the "sinking flux" or "potential" export rather than export. I recommend that the use of the term "export" be reconsidered to see if this accurately reflects what was measured during study.**

Answer :

We agree that the depth of the sediment traps (around 15 m) is too shallow to consider that the sinking matter collected in these traps corresponds to the exported material under the euphotic zone, and this was not our intention. However, guided by a concern for homogeneity throughout the different papers of the VAHINE special issue, we decided to keep the term 'export ' but to add a sentence to make it clear that this term refers to the particulate matter exported at 15 m and not to the particulate matter exported below the euphotic zone :

Section 2.1 : 'Seawater was sampled daily in the three mesocosms (hereafter called M1, M2 and M3) and outside (hereafter called lagoon waters) at three depths (1m, 6m and 12m) and the sediment traps and the material they contain were collected every 24h by scuba divers. It should be noted that the term of 'export' used hereafter does not correspond to the material exported throughout the euphotic zone but to the sinking flux measured in the experiment at 15m depth'.

**Referee #2:**

**Units: Units of nmolN.m-2.d-1 are used and in some sections scientific notation of exponents is incorrect or is not complete (e.g. P12, lines 350 – 356 "5.108 cell.L-1"). The incomplete notation may be a formatting issue that occurred during the proofreading process. Nonetheless, I would advise that SI units are used and suggest that all data in scientific notation is checked that it is correctly printed as this was distracting when reading the manuscript**

Answer :

As far as we understand this comment, we downloaded the .pdf of our manuscript available on the Biogeoscience website to check the notation of exponents and did not see any error in the syntax. The example mentioned above at lines 350-356 appears correctly with '1.5 $10^8$ cell.L$^{-1}$'. We will do our best to check all scientific notations when the revised manuscript is uploaded on the Biogeoscience website.

**SPECIFIC COMMENTS:**

**Referee #2:**

**P1, line 9: It is unclear what kind of P was added during this study. It would be clearer if "P" was changed to "inorganic P" in the abstract that to indicate inorganic P was added to the mesocosms.**

Answer :

The original sentence was replaced by :

'The mesocosms were intentionally enriched with 0.8 $\mu$mol.L$^{-1}$ of inorganic P to trigger..'

**Referee #2:**

**P1, line 13: It is unclear what is meant by population scale here. Please specify.**

Answer :

We agree that the expression 'population scale' was out of context in the Abstract and can be confusing. This expression has to be seen in opposition with the expression 'cellular scale' because we have observed in this study that a time lag was necessary after the DIP enrichment to see a significant development of organisms at the population level.

We propose to rephrase this part of the Abstract as follows:

"5-10 days were necessary to obtain an increase in primary and export productions after the DIP enrichment, thereby suggesting that classical methods (short-term microcosms experiments) used to quantify nutrient limitations of primary production may not be relevant everywhere".

**Referee #2:**

**P2, line 2: "Pacific ocean" should read ". . .Pacific Ocean. . .".**

Answer :

This has been changed.

**Referee #2 :**

**P2, line 41: The bracket before UCYN-A should be a comma i.e. ". . .Group B, UCYN-A . . .".**

Answer :

This has been changed.

**Referee #2 :**

**P3, line 59: It isn't clear what is meant by affected in ". . .a water mass affected by diazotroph development. . .". Please rephrase.**

Answer :

The initial sentence has been replaced by :

"..to isolate a water mass with diazotrophs".

**Referee #2 :**

**P3, line 69: The 15 in "d15N" should be as a superscript.**

Answer :

This has been changed.

**Referee :**

**P3, line 75: A bracket is missing at the end of ". . . Group C (UCYN-C)".**

Answer :

This has been changed.

**Referee #2:**

**P4, line 93: Please specify if "the sediment traps were collected" or instead "the material from the sediment traps were collected".**

Answer :

The material from the sediment traps was collected every 24 hours.

The original sentence has been replaced by :

"..the buckets at the bottom of the sediment traps and the material they contain were collected and replaced by new ones every 24h by scuba divers".

**Referee #2 :**

**P4, lines 112 – 115: Here the sinking velocity was set as constant as 0.7 m/day for the first 10 days but increased according to a polynomial function to up to 10 m/day.**

**-Why was this function used and why would the sinking velocity of the particles suddenly increase after 10 days?**

**- This appears to have a marked and sudden influence on the model output eg. Fig. 7(c), (f) and (m), that does not seem to fit with the experimental data in Fig. 4.**

Answer :

- A polynomial function was used to parametrize the sinking velocity in order to take into account the aggregation process observed in the mesocosms during the second part of the experiment (P2) and the associated increase in the sinking velocity. This function was intended to gradually increase the sinking velocity from day 10, from 0.7 to 10 m/day, without discontinuity. (cf the following figure).

- Concerning the marked and sudden influence in the model output in Fig. 7 mentioned in the referee's comment: First, it must be noticed that these discontinuities do not affect the model variables (see figure 4. (d), (f) and (h) for example), but only the percentage of DDN in the different compartments. Moreover, these discontinuities are not due to an increase in the sinking velocity at day 10, but rather to the set up at day 10 of the sinking of 10 % of the living compartments (phytoplankton, zooplankton and heterotrophic bacteria). Before day 10, only the detrital compartment was allowed to sink in the model. This feature was implemented in order to simulate the aggregation process observed in the mesocosms, which seemed to bring all particles (including

living particles) to the bottom during P2. As a consequence, part of the organisms get out of the system by sinking down to the traps and therefore no longer fill the detrital compartment of the water column. This leads to the abrupt decrease in the DDN proportion in the detritic compartment (Fig. 7, (f)) and to the abrupt increase of this proportion in the traps (Fig 7, (m)). From day 10, dissolved organic carbon (DOC) begins to sink and get out from the mesocosm. As a consequence, the intracellular carbon quota in bacteria ($Q_C$) immediately decreases. Since the hydrolisis of detrital organic matter (DET) into dissolved organic matter (DOM) is regulated in the model by $Q_C$, (i.e. the hydrolysis rate increases when $Q_C$ decreases), the hydrolysis rate increases from day 10 and more DDN passes from $DET_N$ to DON. This is why we observe a slight but sudden increase in DDN in DON on day 10 (Fig. 7 (c)).

[Figure]

**Referee #2:**

**P5, line 130: Language error - should read 'When the total N and P pools (Ntotal and Ptotal) were calculated from the model outputs...'.**

Answer :

This has been changed

**Referee #2:**

**P5, line 153: Is the zooplankton abundance and C, N, P data from this study?**

Answer :

Some counts of zooplankton have been made during the VAHINE experiments but we couldn't use them since they include juveniles (our model only takes into account adults) and there were no data

on their intracellular content in terms of C, N, and P. No intracellular content in C, N or P were directly available from the experiment for Zooplankton, nor for the other organisms included in the model. The intracellular quotas used to convert the cellular abundances into biomasses for the initial conditions were taken from the Eco3M-med model described by Alekseenko et al. (2014). We propose to add this information in section 2.3.1 :

"The initial values of C, N, P and chlorophyll concentrations for the planktonic compartments were derived from the initial cellular abundance data and from their intracellular contents (table 1). These intracellular contents were thus taken from the Eco3M-MED model described in Alekseenko et al. (2014)".

**Referee #2:**

**P5, line 143: Language error – "autotroph phytoplankton" should read "autotrophic phytoplankton".**

Answer :

This has been changed

**Referee #2:**

**P6, lines 167-170: The labile DON fraction is defined as the ". . .quantity consumed during the experiment in the mesocosms which was estimated at 1 umol.L-1". How does this definition fit with the probable production of DON during the study period? Could this labile fraction be underestimated?**

Answer :

We expected a DON increase during the course of the experiment. Finally, only a strong decrease in the DON concentration occurs at the end of the experiment, a process extensively discussed by Berthelot et al. (2015). We agree that the term 'consumed' in line 168 is confusing in the sentence and that the associated explanation is unclear. We propose to rephrase this part as follows :

"To extract the labile fraction from the DON data, we assumed that the plateau reached by the DON concentration at the end of the experiment (4 $\mu$mol.L$^{-1}$) was equal to the concentration of the refractory DON in this study. Considering that the refractory fraction of DON was stable throughout the experiment and fixed at 4 $\mu$mol.L$^{-1}$, from a initial total concentration of 5 $\mu$mol.L$^{-1}$ at day 2, the initial labile fraction was therefore estimated at 1 $\mu$mol.L$^{-1}$".

This hypothesis is entirely consistent with the release and consumption of new DON, a process which is totally taken into account in our model.

Finally, in this paragraph, we only described the initial concentration of DON fixed at 1 $\mu$mol.L$^{-1}$ at $t_0$, which does not imply that the labile DON fraction cannot exceed 1 $\mu$mol.L$^{-1}$ during the study period.

**Referee #2:**

**P8, line 222: The number of cells in a trichome for Trichodesmium sp. is variable rather than a consistent number between trichomes. Hence here, I would suggest using the word "assuming" rather than "considering" here.**

Answer :

The original sentence has been replaced by :

" (assuming that a trichome includes 100 cells.. "

**Referee #2 :**

**P9, line 255: It is unclear what is meant by "diversity parameters". Is this referring to the composition of the plankton community present? Please clarify.**

Answer :

We agree that the word 'diversity' is not suitable here.

This refers to the physical parameters such as temperature, salinity. The original sentence has been replaced by :

"A vertical homogeneity was observed in the mesocosms during the experiment for most of the biogeochemical and physical characteristics"

**Referee #2:**

**P9, line 257 – 261: The distinction between P0, P1 and P2 is clearly defined in the caption of Fig. 3 but not in the body text. It would be helpful to also have this in the text here as well.**

Answer :

The original sentence has been replaced by :

"P0 stands for the few days before the DIP enrichment, P1 is the period when Diatom-diazotroph associations dominate the diazotrophic community (i.e. from day 5 to day 14), and P2 is the period when UCYN-C dominate the diazotrophic community (i.e. from day 15 to day 23)".

**Referee #2:**

**P9, line 273: Should "nmNH4+" instead read "mNH4+"? Is this correct or is this a typing error. Here "remains" should also read "remained".**

Answer :

This is actually a typing error, sorry about that.

This has been corrected to "mNH4+" and to 'remained'

**Referee #2:**

**P10, line 278: The increase in mDOP in SIME described in the body text is very difficult to distinguish from Fig. 3(c). What was the magnitude of this increase?**

Answer :

This is actually a slight increase in DOP from 0.14 to 0.18 $\mu$M. This has been mentioned in the manuscript

The original sentence has been replaced by :

"A slight increase in mDOP in SIME from 0.14 to 0.18 µM and a slight decrease until 0.14 µM and 4.5 µM in mDOP and mDON repsectively are observed during P2".

**Referee #2:**

**- P12, lines 370 – 371: Are the growth rates reported in per second (s-1) as described here in the text? Indeed, these seem very high, with reported rates of over 200 cells L-1 s-1 in Fig. 6. Is this correct? Or is 'x 10-4' missing from the y-axis?**

**- Additionally, it seems that most of the model outputs are quite smooth with minimal variation on a time scale shorter than one day, apart from the turnover time of DIP in the SIME simulation (Fig. 4 (b)).**

**- What is the temporal resolution of the model? Is this consistent across all variables included in the model?**

Answer :

- The specific (per cell) growth rates reported per second in this paragraph are actually correct. There is no missing 'x 10-4' from the y-axis for the Fig.6 (a) concerning the UCYN-C.

The maximum value of 200 $cell.L^{-1}.s^{-1}$ for UCYN-C is consistent with the exponential growth of UCYN-C during P2 in the model outputs (which is overestimated compared to data).

- Diurnal variations are particularly noticeable on variables directly impacted by light such as chlorophyll a, primary production (PP), N2 fixation. However, fluxes such as PP, bacterial production (BP), N2 fixation, were averaged over 24h during the experiment ($\mu molC.L^{-1}.d^{-1}$ for PP and BP, and $nmolN.L^{-1}.d^{-1}$ for N2 fixation) with one dot per day. We chose to represent the model outputs in the same way, therefore concealing the day/night cycle.

The day/night variations are noticeable on the Chla stock (Fig 4. (c)).

Concerning the turnover time of DIP, the variations do not come from the diurnal cycle but from the rapid variations in the P uptake rate by organisms. The latter is constrained by the P intracellular quota through a quota function which is highly non-linear in the vicinity of the maximum quota. Actually, when the P intracellular quota is near to the maximum P quota, the net uptake of P by the organisms decreases quickly which leads to rapid variations in the P intracellular quota when organisms are saturated in P.

- Resolution of the model : 30 seconds.

  Resolution of results recording: 10 min

The temporal resolution of the model is fully consistent across all variables implemented in the model.

**Referee #2:**

**P13, lines 387 – 390: According to Fig. 5(b), UCYN-C was present throughout the study period. Why is the UCYN-C proportion of DDN zero at the beginning of the study period, whereas TRICHO starts with 100%?**

Answer :

At the beginning of the study, the abundances of UCYN-C and TRICHO are quite similar (same order of magnitude) but the N2 fixation flux due to UCYN-C is extremely weak as compared to that ofTRICHO (because TRICHO size is larger than that of UCYN-C and their maximum specific N2 fixation rates are proportional to their intracellular quotas). At t=0 (day 2), the proportion of DDN is zero in every compartment, but after a time step, the proportion of DDN is actually 99.99999 % for TRICHO and 0.00001 % for UCYN-C.

**Referee #2:**

**P14, lines 444-445: "After benefitting diazotrophs, the DDN inputs benefited to nondiazotrophic organisms." – How does the DDN benefit diazotrophs? This could be more clearly and explicitly phrased.**

Answer :

The original sentence has been replaced by :

"After $N_2$ fixation by diazotrophs, the DDN benefitted nondiazotrophic organisms" .

**Referee #2:**

**P14, line 447: The reference to Figure 5(e) appears to be for 4(e) here.**

Answer :

This has been corrected.

**Referee #2:**

**P14, line 452: Here, "synthetized" should read "synthesized".**

Answer :

This has been corrected.

**Referee #2:**

**P15, lines 454 – 456: "As the model does not represent the diatom-diazotroph associations (DDAs), which were the most abundant diazotrophs in the mesocosms during P1 (Turk-Kubo et al., subm., this issue), the modeled export is probably underestimated during P1." From looking at Figure 4 (d), (f), and (h), the modelled data appears to have good agreement with the raw data so I am not sure why the authors suggest that the modelled export is probably underestimated.**

Answer :

We agree with referee #2 that this sentence is confusing. We propose to rephrase it as follows :"Throughout the experiment, Turk-Kubo et al. (2015) have shown that DDA were the most

abundant diazotrophs in the mesocosms during P1. However, it has been shown (Berthelot et al. 2015b) that they did not represent a significant biomass, and the associated export flux was low compared to the export flux measured during P2. Moreover, due to their rapid settling (Villareal et al., 1996), DDN produced by DDA did not benefit the system. For these reasons, we decided not to include DDAs in the model. During P1, the export in SIME is unexpectedly in good agreement with data, likely due to the overestimation of UCYN-C by the model (Fig. 5, (b)). The export during P1 is however lower than during P2, during which we observed a higher increase in detrital particulate matter (Fig. 4, (d), (f) and (h)) enhanced by the UCYN-C growth, not easily noticeable with the log scale on the Fig. 5 (b)".

**Referee #2:**

**P15, line 484: The reference to Figure 5(e) appears to be for 4(e) here.**

Answer :

This has been changed.

**Referee #2:**

**P15, lines 486-487: From Figure 5(b), it appears as though UCYN-C abundances were sampled on day 15, not day 16 as written here.**

Answer :

This has been changed.

**Referee #2:**

**P15, lines 488-489: This question: "Which factor may explain the 10 days delay between the DIP enrichment and the large UCYN-C development?" seems like it should be a sub-heading rather than in the body text.**

Answer :

Thank you for this suggestion, the body text has been reorganized to introduce two sub-headings in this section in order to emphasize this result.

**Referee #2:**

**P17, lines 531-532: Is the study by Van Wambeke et al., subm. From this mesocosm study? If yes, ". . . confirming previous studies . . . " does not accurately reflect this.**

Answer :

We agree this sentence is a bit confusing. We propose to rephrase it in two sentences as follows :

 "Similar results were obtained in the South Pacific gyre for autotrophs (Bonnet et al., 2008) and heterotrophs (Van Wambeke et al., 2008). Van Wambeke et al. (subm, this issue) observed as well, a proximal N limitation of BP at the beginning of the present mesocosm experiment (before the DIP enrichment) at short time scales (1 day or 2 days)".

**Referee #2:**

**P19, line 614: The citation "Verity et al." is missing the year of publication.**

Answer :

The year has been added in the reference

**Referee #2:**

**P19, lines 627 – 629: "Finally, the overestimation of UCYN-C abundance by the model also supports the idea that UCYN-C sinking is underestimated by the model". Could underestimated grazing rates also explain this overestimation of UCYN-C abundances in the model?**

Answer :

This could be actually another explanation for the overestimation of UCYN-C abundances in the model. However, as we do not have enough observations regarding the three zooplankton groups implemented in the model (HNF, CIL, COP), we do not have the relevant information to validate this assumption. This is why we favoured the underestimation of the UCYN-C sinking in the model as an explanation for the overestimation of UCYN-C during P2. This is consistent with the agregation phenomenon which occured in the mesocosms during P2 and which was not well understood during this study. We therefore propose to add the following sentence to the full text, as advised by referee #2 :

"The overestimation of UCYN-C in the model during P2 might also be explained by an underestimation of the grazing by HNF and CIL. Nevertheless, we did not go further in this assumption since few data regarding grazing rates by zooplankton were available in this study".»

**Referee #2:**

**P20, line 650: This observation of the majority of DDN in the DON and NH4+ pools from the model output is in good agreement with observations from experimental work which is cited in the introduction (P2, line 49). These observations would also fit nicely to the conclusions drawn from this modelling study but are not cited in the context of the discussion here.**

Answer :

Thank you for this suggestion, we propose to add the following sentence in the section 4.3 :

"Finally, though DDN transits in the same proportions in NH4 and DON, it mostly accumulates in DON since DDN-NH4 is taken up more rapidly, these results substantiating those found by Berthelot et al. (2015a)".

**Referee #2:**

**Figure 1: Does the model include mortality of all diazotrophs or just Trichodesmium sp.? Is the arrow from N2 to DOM via mineralisation included in the model? If yes, is this mediated by diazotrophs or is this a separate process as indicated in this figure? How is the uptake of**

**DOP by diazotrophs incorporated in the model? Typing errors: "morality" should be "mortality" and "Smal" should be "Small".**

Answer :

Mortality is for both Trichodesmium and UCYN-C.

The mineralisation arrow on the left starts from the DET compartment to the DOM. There is no link with 'photosynthesis' or 'N2'.

The uptake of DOP by diazotrophs is incorporated in the model for diazotrophs but regulated by an inhibition function which allows the DOP uptake only when the DIP concentration is low.

The figure has been partially redrawn to avoid confusion and the errors mentioned have been corrected.

**Referee #2:**

**Figure 3: The y-axis label of PO43- in Figure 3 is not consistent with the figure caption which abbreviates phosphate to DIP.**

Answer :

This has been changed.

**Referee #2:**

**Table 1: How was the living fraction of POM (i.e. "POC-living") calculated?**

Answer :

The living fraction of POM in terms of C, N or P is the sum of each living compartment in terms of C, N or P. This is easy to calculate in the model.

**Referee #2:**

**Table 2: The superscripts on NO3, NH4 and PO4 should instead be subscripts in both the "Parameter" and "Definition" columns. "Exsudation" should also read "Exudation". I would recommend adding "cell" to the "minimum quota of . . ." to read "minimum cell quota of . . .".**

Answer :

These changes have been made.

**Anonymous Referee #3**

**GENERAL COMMENTS :**

**Referee #3:**

**10-day delay: This is an important result suggesting that there is a 10-day lag between the phosphate enrichment and the increase in diazotrophs. First, it is often confused in the text**

the response between nitrogen fixation and diazotroph concentration, which needs to be corrected. It is not diazotrophs but nitrogen fixation that responds with a delay. Figure 5 shows that Trichosdesmium and UCYNC increase concentration right after the phosphate addition (when looking at model results).

Answer :

Actually, the UCYN-C concentration increases clearly 10 days after the enrichment by a peak of abundances around day 20 in the model, but this result does not clearly appear in the figure because of the log scale. This feature is however noticeable in figure 6 (a) where the growth rate of UCYN-C reaches 200 cell.s$^{-1}$ at the end of P2. It is true that the development of UCYN-C began right after the DIP enrichment, but the significant growth of the UCYN-C population occurs later during P2 with a lag of 10 days after the enrichment. The following figure shows the UCYN-C abundances with a linear scale emphasizing their strong development during P2.

[Figure]

Referee #3:

**In addition if the authors base their results on nitrogen fixation rate, there is the issue that the model does not match the observations for nitrogen fixation: observed nitrogen fixation rises 5 days after the enrichment (day 10, based on Figure 5), whereas modelled nitrogen fixation rises 10 days after.**

Answer :

We agree that the increase in N2 fixation in the model is less sudden than in the data. However, even if we see a slight increase in the observed N2 fixation rates from days 11 to 13 (with two

extreme dots for M1 and M3), the significant increase in the N2 fixation rates begins day 15 for the 3 mesocosms.

**Referee #3:**

**To conclude that there is a 10-day delay is quite an uncertain result, which needs to be clearly mentioned and discussed. Finally, the text does not describe accurately the results in the model and data: "Since we observed the increase in PP and BP only after 10 days in both experimental and simulation results" (line 537). This is not what Figure 4 shows and should be amend accordingly as well as tone down the conclusion "we may conclude that 10 days are necessary for the newly fixed N by diazotrophs to sustain the observed high production rates, and to see an effective change in the planktonic populations".**

Answer :

We agree that the '10-day delay' is quite restrictive as there are actually some differences in the delay. In Figure 4 (e) and (g), except for Mesocosm 1 which presents a steady increase in PP and BP from day 10 to the end of the experiment, data observed in mesocosms 2 and 3 show a more significant increase in PP and BP at day 15 than at day 10. We can therefore rephrase the sentence line 537 as follows :

"Since we observed a significant increase in PP and BP about 10 days after the DIP enrichment in both experimental (M2 and M3, 5 days for M1) and simulation results, we may conclude that 5-10 days are necessary for the newly fixed N by diazotrophs to sustain the observed high production rates, and to see an actual change in the planktonic populations".

**Referee :**

**\*UCYN as the dominant nitrogen fixers: It is not clear from the presented model results (mainly figure 5) that UCYN-C is the dominant nitrogen fixers. I agree that the increase is stronger for UCYN than Trichodesmium, but there isn't enough presented evidence that UCYN explains most of the nitrogen fixation increase (especially considering that Trichodesmium have a larger volume than UCYN). Can you provide more evidence to validate this important point?**

Answer :

The role of UCYN-C regarding the nitrogen fixation has been highlighted by Bonnet et al. (in review), who have shown during an *in vivo* experiment that UCYN-C accounted for $90 \pm 29$ % of bulk $N_2$ fixation during P2.

**Referee #3:**

**\*Length of the manuscript: While the structure of the manuscript is good (logical, wellorganised), the sections are overall too long with too many not necessary relevant details.**

**This makes the manuscript difficult to follow. I would encourage the authors to shorten the paper as much as possible while staying precise and concise.**

Answer :

Thank you for this suggestion, it has been taken into account in the revised manuscript.

**SPECIFIC COMMENTS :**

**Referee #3:**

**- The 1D model has 14 levels of 1 m each, so represents only the first 14 m of the water column. This seems quite shallow as most places the mixed layer is 50-200 m deep. Can you justify this choice?**

Answer :

The aim of this study was to propose a modelling approach of the VAHINE mesocosm experiment. This is a 14m-1D model to be consistent with the height of the mesocosms (which were 14m height).

**Referee #3:**

**- There are too many acronyms making the reading quite difficult. Can you use the full name at the start of a new section at least on the key messages ?**

Answer :

This suggestion has been taken into account in the revised manuscript

**Referee #3:**

**- How do you know that phosphate is not used directly by other organisms and that the increase in PP, Chla is due to DDN?**

Answer :

All the photosynthetic organisms and heterotrophic bacteria can directly consume DIP in our model. Because of the DIP enrichment, organisms are no longer limited by phosphate, but they remain N-limited, and the only external nitrogen source is provided by diazotrophy (see Berthelot et al., 2016). The increase in primary production during P2 is therefore necessarily due to the transfer of DDN from diazotroph to non-diazotroph organisms, except at the end of the experiment when DON consumption represents a significant additional source of N (Berthelot et al. 2016).

**Referee #3:**

**Part 4.1: Mismatch between the description of model results and observations**

Answer :

We agree that this section could be more precise concerning the description of the model results and/or the observations. Some sentences in this section have been rephrased in the revised manuscript in order to clarify the discussion.

**- Figure 6: Units overlapping**

Answer :

The layout in figure 6 has been changed.

**Referee #3:**

**- Figure 6 ? Why if the population growth rate start only at day 5 but UCYN-C are growing earlier?**

Answer :

As far as we understand the question, referee #3 is asking why the population growth rate increases dramatically 5 days after the DIP enrichment (i.e. around day 10) in figure 6 (a), whereas the abundances of UCYN-C in figure 5 (b) begin to increase from the day of the DIP enrichment.

We think that this is again a question of scale, as already mentioned in a previous answer. We agree that with the log scale in figure 5 (b), the dramatic increase in UCYN-C abundances during P2 is reduced, which seems to suggest that the development of the UCYN-C population is constant since the DIP enrichment. However, if we look at the UCYN-C abundances with a linear scale, we can see that the increase in the population growth rate of UCYN-C becomes significant during P2.

[Figure]

[Figure]

**Referee #3:**

**- Why not include DDA in the model since it looks like they play a big part in the first part of the experiment?**

Answer :

We agree that Turk-Kubo et al. (2015) have shown that DDA were the dominant diazotroph organisms during the first part of the experiment compared to the entire diazotroph community.

However, it seems that even if DDAs dominated the diazotroph community during P1, they did not really impact the ecosystem in terms of Nitrogen source because they are rapidly exported (Berthelot et al. 2016). Actually, DDAs have a high sinking capacity, which prevents them from reaching high growth rates. Since this modelling study was focused on the fate of DDN in the planktonic food web, it did not seem relevant to introduce additional complexity and uncertainty in the model by the addition of this compartment, which does not contribute significantly to the transfer of DDN in the food web. We propose to rephrase a paragraph in section 4.1 as follows :

"Throughout the experiment, Turk-Kubo et al. (2015) have shown that DDA were the most abundant diazotrophs in the mesocosms during P1. However, it has been shown (Berthelot et al. 2015b) that they did not represent a significant biomass, and the associated export flux was low compared to the export flux measured during P2. Moreover, due to their rapid settling (Villareal et al., 1996), DDN produced by DDA did not benefit the system. For these reasons, we decided not to include DDAs in the model. During P1, the export in SIME is unexpectedly in good agreement with data, likely due to the overestimation of UCYN-C by the model (Fig. 5, (b)). The export during P1 is however lower than during P2, during which we observed a higher increase in suspended particulate matter (Fig. 4, (d), (f) and (h)) enhanced by the UCYN-C growth, not easily noticeable with the log scale on the Fig. 5 (b)".

**Referee #3:**

**- Can you justify the location of the study in the abstract?**

Answer :

We propose to rephrase the first lines of the abstract as follows :

"The VAHINE mesocosm experiment took place in the oligotrophic waters of the Noumea lagoon (New Caledonia), where high $N_2$ fixation rates and abundant diazotroph organisms were observed. This experiment aimed to assess the role of the nitrogen input through $N_2$ fixation on carbon production and export, and to study the fate of diazotroph-derived nitrogen (DDN) throughout the planktonic food web".

**Referee #3:**

**- Line 34: Tone down the "only few" as there are quite a few models now that incorporate state variables for nitrogen fixers. Add some more references.**

Answer :

This sentence aims to introduce the different references cited in the introduction on the diazotrophy process in models. We agree that the reference of Rabouille et al. (2006) at the end of the sentence is quite confusing.

We propose to rephrase this section as follows :

"Biogeochemical models including N2 fixation have been developed over the last decades, some of

them including diazotrophic organisms as state variables, as described here. In these models, Trichodesmium is the most frequently represented organism since it is the most widely studied diazotroph and its physiology is well documented in the literature (Moore et al., 2001b; Fennel et al., 2001; Moore et al., 2004; Rabouille et al., 2006). In recent studies, other diazotrophs such as unicellular cyanobacteria (termed UCYN) or diatom-diazotroph associations (termed DDA) have been implemented in biogeochemical models".

**Referee #3:**

**- Line 43: also DDA were modelled in Monteiro et al. (2010, 2011)**

Answer :

We acknowledge that we should have cited these papers. This has been done in the revised manuscript, as follows :

"More recently, other biogeochemical models, including a more complex planktonic food web and the contribution of *Trichodesmium* sp., DDAs, UCYN-A and UCYN-B, have been developed (Monteiro et al., 2010, 2011), together with models representing Trichodesmium sp., and a general group of UCYN (Dutkiewicz et al., 2012)".

**Referee #3:**

**- Line 100: Can you say which boundary conditions were implemented for Eco3M?**

Answer :

There are no boundary conditions in our model as we implemented a 1DV box-model without an explicit representation of physical processes. The only physical process lies in the sinking of material (in other words, the sinking flux is the only mass exchange between the 1m height boxes of the model, these boxes being considered homogeneous).

**Referee #3:**

**- Section 3.1 is a good description of the results, but it needs to be specified more clearly if they are model or data results.**

Answer :

Thank you for this suggestion, the section has been reviewed and clarified concerning the origin of results (model or data results).

**Referee #3:**

**- Line 370: Can you define more what the difference is between specific growth and population growth as important later on to describe the model results?**

Answer :

We propose to rephrase line 370 :

"The population growth rate (in $cell.L^{-1}.s^{-1}$), for TRI and UCYN-C, as well as the specific (i.e. per

cell) growth rate (in $s^{-1}$) of TRI and UCYN-C are plotted in Fig. 6 (c) and (d)".

**Referee #3:**

**- Line 378: Define fgrowthTRItrich and fgrowthTri**

Answer :

These two terms are indeed unnecessarily complicating the text and have been removed.

**Referee #3:**

**- Figure 7:**

**-Is that the correct initial condition in the sense that there are more TRI than UCYNC?**

**-Why TRI never come back even after the PO4 enrichment.**

**-Do observations show higher TRI concentration than UCYN?**

Answer :

-We understand that referee #3 is asking why the initial proportion of DDN is up to 100 % in TRI according to Figure 7 (a).

At the beginning of the study, the abundances of UCYN-C and TRICHO are quite similar (same order of magnitude), but the N2 fixation due to UCYN-C is extremely weak as compared to that of TRICHO (because TRICHO size is larger than that of UCYN-C and their maximum specific N2 fixation rates are proportional to their intracellular quotas). At t=0 (day 2), the proportion of DDN is zero in every compartment, but after a time step, the proportion of DDN is actually 99.99999 % for TRICHO and 0.00001 % for UCYN-C.

- Concerning the decline of TRI throughout the simulation, this is due to their low growth rate (0,17 $d^a$). The balance between their growth and their mortality in the model does not allow them to grow significantly over the 25 days of the simulation.

This assumption seems to be consistent with data as no bloom of trichodesmium was observed in the mesocosms.

- Observations do not show higher TRI concentration than UCYN during the experiment. UCYN-C indeed dominate considerably the diazotroph community from day 15 until the end of the experiment (Turk-Kubo et al. 2016).

**Referee #3:**

**Line 423: It is not obvious from the observations (Figure 4a, black diamonds) that there is a decrease in nitrogen fixation. Can you please explain/amend?**

Answer :

Between the beginning and the end of P1, N2 fixation in surrounding waters decreases from 10 $nmolN.L^{-1}.d^{-1}$ to 5 $nmolN.L^{-1}.d^{-1}$. We acknowledge that there are slight variations (e.g. the small peak at day 12) but the general trend is nevertheless decreasing.

**Referee #3:**

**- Line 439: "the DIP enrichment resulted in an increase in the abundances of all planktonic groups except PHYL and COP". This is not what the observations show for PHYL.**

Answer :

This sentence concerns the model results, which are indeed not very clear here.

We propose to complete the sentence as follows :

"the DIP enrichment resulted in an increase in the abundances of all planktonic groups in the model outputs except PHYL and COP" .

**Referee #3:**

**- Looking at Figure 4, the system looks like it is loosing DIP during the standard model run, a trend not present in the observations. Have you thought about this feature and do you think it might impact your model results (for ex, in relation to TRI, PHYL and COP)?**

Answer :

We understand that referee #3 notices the decrease in TDIP in SIM_C (without the DIP enrichment) below the values observed in the mescosms data (black diamonds) in Fig. 4 (b). Actually, this difference between the model outputs and the experimental data is negligible, but is enhanced by the log scale of this figure. Here is the figure of TDIP represented with the linear scale :

[Figure]

**References :**

-Alekseenko, E., Raybaud, V., Espinasse, B., Carlotti, F., Queguiner, B., Thouvenin, B., Garreau, P., and Baklouti, M.: Seasonal dynamics and stoichiometry of the planktonic community in the NW Mediterranean Sea; a 3D modeling approach, Ocean Dynamics, 64, 179–207, 2014

-Berthelot, H., Moutin, T., L'Helguen, S., Leblanc, K., Hélias, S., Grosso, O., Leblond, N., Charrière, B., and Bonnet, S.: Dinitrogen fixation and dissolved organic nitrogen fueled primary production and particulate export during the VAHINE mesocosms experiment (New Caledonia lagoon), Biogeosciences Discussions, 12, 4273–4313, doi:10.5194/bgd-12-4273-2015, 2016.

- S. Bonnet, H. Berthelot, K. Turk-Kubo, S. Fawcett, E. Rahav, S. l'Helguen, and I. Berman-Frank, Dynamics of $N_2$ fixation and fate of diazotroph-derived nitrogen in a low nutrient low chlorophyll ecosystem: results from the VAHINE mesocosm experiment (New Caledonia). Biogeosciences Discuss., 12, 19579-19626, doi:10.5194/bgd-12-19579-2015, in review

- Turk-Kubo, K., Frank, I., Hogan, M., Desnues, A., Bonnet, S., and Zehr, J. P.: Diazotroph community succession during the VAHINE mesocosms experiment (New Caledonia Lagoon), Biogeosciences, subm.

---

## Referee Report (RR1)

**'Biogeochemical fluxes and fate of diazotroph derived nitrogen in the food web after a phosphate enrichment: Modeling of the VAHINE mesocosms experiment'**

Gimenez et al.

**Referee comments**

The manuscript has improved considerably in response to the referee comments during revision, hence I recommend publication of this manuscript by Gimenez et al. in Biogeosciences after the following technical revisions:

- Correction of scientific notation from 1.5 $10^{-4}$ to 1.5 x $10^{-4}$.
- Correction of stable isotope ratio notation to $\delta^{15}N$ consistently throughout manuscript.
- Revision and reduction of use of brackets, particularly for citations and those including 'this issue', to improve readability.
- Revision of citations for 'this issue' for consistency throughout manuscript as some cited manuscripts from this Special Issue were not identified as such.
- Revision of references section to ensure italics and sub/superscripts are correct e.g. 'N$_2$', '15N2', and 'O2'.
- Language has improved considerably in the revised manuscript however some minor errors are still present hence proof-reading is recommend by authors (unless this will be done by BG during publication process).
- P3, line 65: It is unclear what is meant by 'diversity' here and this term should be revised/clarified (see also comments from Referee #2 on BGD manuscript).
- P6, lines 192 – 194: Addition of citation for the choice of ratios between BAC/HNF/CIL abundances (i.e. HNFcell, BACcell, CILcell).